# The winter 2019 air pollution (PM$_{2.5}$) measurement campaign in Christchurch, New Zealand

Ethan R. Dale[1], Stefanie Kremser[1], Jordis S. Tradowsky[1], Greg E. Bodeker[1], Leroy J. Bird[1], Gustavo Olivares[2], Guy Coulson[2], Elizabeth Somervell[2], Woodrow Pattinson[†], Jonathan Barte[3], Jan-Niklas Schmidt[4], Nariefa Abrahim[5], Adrian J. McDonald[6], and Peter Kuma[6]

[1]Bodeker Scientific, 42 Russell Street, Bridge Hill, Alexandra 9320, New Zealand.
[2]National Institute of Water and Atmospheric Research (NIWA), 41 Market Place, Auckland Central 1010, Auckland, New Zealand.
[3]Météo-France, 42 avenue Gaspard Coriolis, 31100 Toulouse, France.
[4]Luisental 28, 28359 Bremen, Germany
[5]University of Otago, 362 Leith Street, North Dunedin, Dunedin 9016, New Zealand.
[6]University of Canterbury, 20 Kirkwood Avenue, Upper Riccarton, Christchurch 8041, New Zealand.
[†]Deceased 15 March 2020.

**Correspondence:** Ethan Dale (ethan@bodekerscientific.com)

**Abstract.** MAPM (Mapping Air Pollution eMissions) is a project whose goal is to develop a method to infer airborne particulate matter (PM) emissions maps from in situ PM concentration measurements. In support of MAPM, a winter field campaign was conducted in New Zealand in 2019 (June to September) to obtain the measurements required to test and validate the MAPM methodology. Two different types of instruments measuring PM were deployed: ES-642 remote dust monitors (17 instruments) and Outdoor Dust Information Nodes (ODINs; 50 instruments). The measurement campaign was bracketed by two intercomparisons where all instruments were co-located, with a permanently installed Tapered Element Oscillating Membrane (TEOM) instrument, to determine any instrument biases. Changes in biases between the pre- and post-campaign intercomparisons were used to determine instrument drift over the campaign period. Once deployed, each ES-642 was co-located with an ODIN. In addition to the PM measurements, meteorological variables (temperature, pressure, wind speed and wind direction) were measured at three automatic weather station (AWS) sites established as part of the campaign, with additional data being sourced from 27 further AWSs operated by other agencies. Vertical profile measurements were made with 12 radiosondes during two 24-hour periods and complimented measurements made with a mini micropulse lidar and ceilometer. Here we present the data collected during the campaign and discuss the correction of the measurements made by various PM instruments. We find that when compared to measurements made with a simple linear correction, a correction based on environmental conditions improves the quality of measurements retrieved from ODINs but results in over-fitting and increases the uncertainties when applied to the more sophisticated ES-642 instruments. We also compare PM$_{2.5}$ and PM$_{10}$ measured by ODINs which, in some cases, allows us to identify PM from natural and anthropogenic sources. The PM data collected during the campaign are publicly available from https://doi.org/10.5281/zenodo.4542559 (Dale et al., 2020b), the data from other instruments are available from https://doi.org/10.5281/zenodo.4536640 (Dale et al., 2020a).

## 1 Introduction

Airborne particulate matter (PM) comprises particles that can be solid, liquid or a mixture of both. The solids comprising PM can include both organic and inorganic constituents, such as sea salt, dust, pollen, and soot. Particle sizes and composition vary with location, origin and in situ chemical processes (Adams et al., 2015). There are health concerns associated with PM
emissions, as PM remains suspended in the air where, if it is inhaled, the risk of developing cardiovascular and lung-related diseases increases (Anderson et al., 2012; Pizzorno and Crinnion, 2017). The World Health Organization estimates that PM air pollution contributes to approximately 800,000 premature deaths each year, ranking it the 13[th] leading cause of mortality globally (Anderson et al., 2012). Pope et al. (2009) show that by decreasing the ambient $PM_{2.5}$ concentration by 10 $\mu gm^{-3}$ life expectancy can be increased by 0.6 years. PM can be described by its aerodynamic equivalent diameter (AED) and particles
are generally subdivided according to their size: $< 10, < 2.5$, and $< 1$ μm ($PM_{10}$, $PM_{2.5}$, and $PM_1$, respectively). Particles with a diameter greater than 10 μm have a relatively small suspension half-life and are largely filtered out by the nose and upper airway if inhaled. Particles with diameters between 10 and 2.5 μm ($PM_{10-2.5}$) are referred to as 'coarse', less than 2.5 μm as 'fine', and less than 1 μm as 'ultrafine' particles. It is important to note that $PM_{10}$ encompasses ultrafine ($PM_1$), fine ($PM_{2.5-1}$), and coarse ($PM_{10-2.5}$) fractions.
During winter, towns and cities in New Zealand suffer from elevated levels of PM primarily resulting from the burning of wood and coal for home heating (Ministry for the Environment & Stats NZ, 2018). Poor air quality is a more frequent problem in cities and towns that are located in the South Island. This reflects the climatologically colder winters, that occur in the South Island, resulting in greater use of solid fuel for home heating and the formation of capped boundary layers that restrict the dispersion of pollutants being more likely. This study presents measurements of PM made during a winter field campaign in
Christchurch in 2019.

To provide the regional government responsible for managing emissions of PM with legislative tools to address poor air quality, the New Zealand government defined national environmental standards (hereafter NES) for air quality in 2004 and updated these in 2011. The standards include five main air contaminants, viz. $PM_{10}$, sulphur dioxide ($SO_2$), carbon monoxide (CO), nitrogen dioxide ($NO_2$), and ozone ($O_3$). Each contaminant is monitored in 89 geographical regions surrounding urban
areas known as airsheds, Christchurch lies within a single airshed (Fig. 2). Within each identified airshed, a limited number of $PM_{10}$ exceedances of a daily mean limit of 50 $\mu gm^{-3}$ are permitted each year (one for some airsheds, three for others). However, the PM standard is currently under review with the expectation that the primary standard for PM pollution will shift from $PM_{10}$ to $PM_{2.5}$ in recognition of $PM_{2.5}$ being more relevant for assessing health impacts, since it penetrates deeper into the lungs than $PM_{10}$. This proposed change will bring New Zealand's air quality standards in line with those suggested by the
World Health Organization (WHO Regional Office for Europe, 2017). As such, while $PM_{10}$, $PM_{2.5}$ and $PM_1$ were measured during the field campaign, this paper focuses primarily on $PM_{2.5}$.

## 1.1 The Mapping Air Pollution eMissions (MAPM) project

The goal of the MAPM project, funded through the New Zealand Ministry of Business, Innovation and Employment, is to develop a method for inferring daily, high spatial resolution ($< 100$ m) $PM_{2.5}$ emissions maps for cities. The MAPM method uses an inverse model that takes as input in situ $PM_{2.5}$ mass concentration measurements and the meteorological data required to calculate trajectories from sources to receptors (instrument locations) and generates $PM_{2.5}$ emissions maps and their uncertainties (hereafter referred to as 'the MAPM methodology'). Several linked lines of development, conducted in parallel, form the basis of the MAPM research:

1. A field campaign to generate the data required to test and validate the MAPM methodology.

2. A forward model that simulates the local meteorology over the duration of the campaign. This model is used to drive Lagrangian particle dispersion trajectories and produce source-receptor relationships between the $PM_{2.5}$ sensors and the emissions sources.

3. An inverse model that takes the source-receptor relationships, in situ $PM_{2.5}$ concentration measurements and a prior emissions map as input to generate daily maps of sources of $PM_{2.5}$ emissions and their uncertainties.

4. Several Observing System Simulation Experiments that are being used to explore the effects of different (i) instrument configurations, and (ii) instrument types and associated measurement uncertainties.

Because MAPM's purpose is to infer $PM_{2.5}$ emissions maps for cities, Christchurch was selected as a target to demonstrate MAPM's capability, as it is one of the largest cities in New Zealand and PM concentrations in Christchurch frequently exceed the NES thresholds during winter. As a result, a three month measurement campaign was conducted in Christchurch in 2019, which provides the required $PM_{2.5}$ measurements that are used as input to the inverse model, which is used to infer PM emissions sources in Christchurch. This paper describes this field campaign and obtained measurements in detail. For a detailed description about the inverse model and inferred emissions maps, the reader is referred to Nathan et al. (2021).

## 1.2 Previous PM measurement field campaigns conducted in Christchurch

In addition to the three PM permanent measurement sites that are installed for regulatory purposes in Christchurch, there have been several previous short-term PM measurement campaigns in Christchurch and surrounding areas. During the winter of 2016, 19 ES-642 remote dust monitors (hereafter referred to as ES-642), measuring both $PM_{10}$ and $PM_{2.5}$ were deployed across the Christchurch airshed. This network was designed to have a high level of correlation with permanent reference instruments operated by Environment Canterbury (ECan) and primarily focused on suburban PM concentrations, with some information from local emissions.

Between May and November 2017 an additional 10 low cost nephelometers units were deployed to focus on denser measurement networks to investigate the prevalence of spikes and airshed boundary gradients using the 2016 spatial characterisation of the airshed. Both the 2017 and the 2016 campaign found significant spatial and peak PM differences to the data from the 3 permanent monitoring sites.

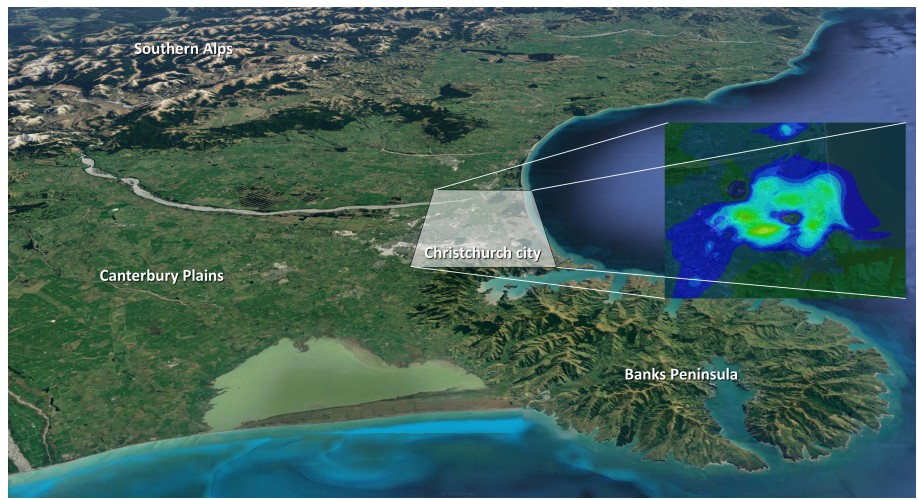

**Figure 1.** The geographical context for Christchurch showing the Southern Alps to the west, Banks Peninsula to the east, and the Canterbury Plains between the city and the Southern Alps. The inset shows a typical PM$_{2.5}$ distribution around the city. Background image: © Google, Maxar Technologies.

Within MAPM, the measurements from the 2016 and 2017 measurement campaigns were combined using a regression
model to create high resolution hourly PM$_{2.5}$ maps for Christchurch, which were then used as input to an algorithm that selected locations for the placements of Outdoor Dust Information Node (ODIN) and ES-642 instruments for the 2019 campaign (refer to Sect. 3).

Another measurement campaign was undertaken in autumn 2016 by Huggard et al. (2019). 18 ODIN nephelometers were installed in Rangiora, a small town $20\ \mathrm{km}$ north of Christchurch. Data from these were compared to measurements made by
a permanent TEOM also installed in Rangiora. Huggard et al. (2019) analysed several methods of correcting ODIN PM data against a TEOM reference. They found little benefit in increasing the instrument co-location period beyond seven days and that a correction based on relative humidity was optimal.

## 1.3 Description of Christchurch meteorology and sources of particulate matter

Christchurch is the main urban centre of the Canterbury region, which is situated on the east coast of New Zealand's South
Island. It is located on the eastern fringe of the Canterbury Plains that slope gently from the coast to the Southern Alps that rise to elevations well above 3000 m. While Christchurch is situated on generally flat terrain, immediately south of the main urban area, the Port Hills form the northernmost side of the volcanic landscape of Banks Peninsula, provide a local orographic feature that reaches elevations of up to 450 m (Fig. 1).

Dwellings in the urban area of Christchurch are mainly single story houses and buildings higher than 5 stories are rare in the
city centre. The current tallest building in Christchurch rises to 86 metres. Many of the high-rise buildings were demolished following a series of major earthquakes in 2010 and 2011. Christchurch has a relatively low population density ($270\ \mathrm{km}^{-2}$

compared to $1,510\ \text{km}^{-2}$ for London, UK). In the centre of Christchurch is Hagley park with an area of $1.65\ \text{km}^2$ in this area, very little PM emissions occur.

Christchurch has a temperate maritime climate with warm dry summers and winters in which it is common for temperatures to fall below $0°\text{C}$ overnight. There are, on average, 70 days of ground frost per year. Snowfalls occur on average once or twice a year on the Port Hills and about once every two years on the plains. The dominant topography that modifies the synoptic flow around Christchurch are the Southern Alps which form a roughly perpendicular obstacle to the predominant westerly wind. The resultant foehn-type winds lead to Christchurch having relatively low rates of rainfall that limit rainout of airborne PM pollution. The second most common wind in Christchurch is an onshore easterly wind that flows parallel to the Port Hills, which also induces the majority of the rainfall.

During winter, the main source of $PM_{2.5}$ emissions in Christchurch is burning wood and coal for home heating. Further minor anthropogenic sources result from industry and transport with natural sources including dust and sea salt. ECan monitors $PM_{10}$ at two locations in Christchurch (Woolston and St Albans) to provide the data needed to detect exceedances of the NES permitted thresholds. High pollution days can often be related to several precursor states occurring in concert such as meteorological conditions, topography influencing air mass movement, and short-term emission sources such as passing heavy or poorly serviced vehicles (Mukherjee and Toohey, 2016).

In 2019, Christchurch reported seven days where the daily mean $PM_{10}$ concentration exceeded the $50\ \mu\text{gm}^{-3}$ NES permitted threshold (i.e. four days more than is currently permitted; from 1 September 2020, only a single exceedance is permitted each year). The proposed new limits for any airshed are: (i) no more than three exceedances of $25\ \mu\text{gm}^{-3}$ for daily mean $PM_{2.5}$ and (ii) an annual mean $PM_{2.5}$ concentration of no more than $10\ \mu\text{gm}^{-3}$. During winter, 90 % of all particulates measured as $PM_{10}$ comprise particles smaller than $2.5\ \mu\text{m}$ (Aberkane et al., 2010). A series of major earthquakes occurred in 2010 and 2011 in Christchurch, resulting in major structural damage, which substantially increased the reliance on woodburning for home heating. This, together with intensive construction and demolition activities elevated several sources of PM pollution (Tunno et al., 2019). On the other hand, major damage led to many homes being removed, people moving away and, older wood burners being replaced with lower emission burners or electrical heating, leading to reduced PM emissions.

Sources of PM in Christchurch's surrounding areas include agricultural fires and agricultural dust, as well as sea salt from the nearby ocean. Agricultural fires occur predominantly between February and March and are often forbidden during summer for safety reasons. Golders Associates (2014) investigated the impact of burning of crop residue and found that while agricultural fires were not likely to cause an exceedance of the NES, large spikes in $PM_{10}$ were possible at hourly timescales and recommended that agricultural fires are not burned within $6\ \text{km}$ of an urban area.

This paper describes each of the instruments used in the campaign (Sect. 2), the algorithm used to decide where to locate the sensors (Sect. 3), how the sensors were inter-calibrated and the QA/QC (Quality Assurance/Quality Control; Sect. 4), the method used to derive the uncertainties on the $PM_{2.5}$ measurements (Sect. 5), with a final description and presentation of the data in Sect. 6. Concluding remarks regarding the intended use of the data are provided in Sect. 7.

## 2  Instruments

The MAPM field campaign was conducted in Christchurch from 4 June to 9 September 2019 to collect PM concentration and meteorological measurements. The campaign was made up of two co-location periods (6-12 June and 30 August to 8 September) which bracketed the main deployment period (22 June to 25 August). Data from the co-location periods, where all PM instruments were installed alongside each other was used for the correction of measurements (Sect. 4), during the deployment period instruments were distributed across the city. 50 ODIN and 17 ES-642 instruments were distributed throughout the city, measuring PM concentration every minute at ground level (i.e. around 2 to 3 m above the surface depending on the instrument type). Three automatic weather stations (AWS) that measured temperature, humidity, wind speed, and wind direction were installed at the perimeter of the city (Fig. 2). Measurements from these AWSs were complemented by measurements from AWSs operated by the Meteorological Service of New Zealand (MetService) and the National Institute of Water and Atmospheric Research (NIWA), as well as meteorological measurements made by the public and submitted to the United Kingdom Met Office weather observation website (WOW; https://wow.metoffice.gov.uk/). A micropulse lidar and a ceilometer installed on top of a building (45 m altitude above surface) measured vertical profiles of aerosol concentration. To investigate the stability of the boundary layer, its height, and to identify the occurrences of temperature inversions, 12 balloon-borne radiosondes were also deployed during the field campaign.

## 2.1  ES-642 remote dust monitor

The ES-642, produced by Met One Instruments, Inc., is a type of nephelometer which automatically measures real-time airborne particulate matter concentrations using the principle of forward laser light scatter. The sensor has a prescribed accuracy of $\pm 5$ % and a sensitivity of $1\ \mu gm^{-3}$ (Met One Instruments, Inc, 2019). Air is drawn into the sensor through a sharp-cut cyclone to prevent particles larger than $2.5\ \mu m$ entering the sensor. The accuracy of a nephelometer is hindered by water vapour present within the sample air. As relative humidity increases above 50 % particles begin to aggregate and increase in size due to water absorption (Di Antonio et al., 2018). To mitigate these effects, a 10 W inlet heater is used to warm the incoming air and thereby lowering the relative humidity of the air entering the sensor, preventing the intake of water vapour. The heater turns on when the ambient relative humidity reaches values above 40 %. The sampled air then passes through the laser optical module where the suspended particles in the air stream scatter the laser light through reflective and refractive properties. This scattered light is collected onto a photodiode detector at a near-forward angle, and the resulting electronic signal is processed to derive a continuous, real-time measurement of airborne PM concentrations.

The ES-642 instruments were provided by MOTE Ltd. and were coupled with data modems to transmit data in near real-time. The instruments were deployed in two different configurations (referred to collectively as ES-642s hereafter): 'Dust Motes' (DM) consisting of a ES-642 module and 'Dust Met Motes' (DMM) consisting of a ES-642 module and a sonic anemometer which measures the airflow in the vicinity of the instrument.

Nine Dust Motes and five Dust Met Motes were deployed throughout Christchurch during the MAPM field campaign (Fig. 2). A further three ES-642s are permanently installed and operated in Christchurch by ECan. Thus, 17 ES-642s were

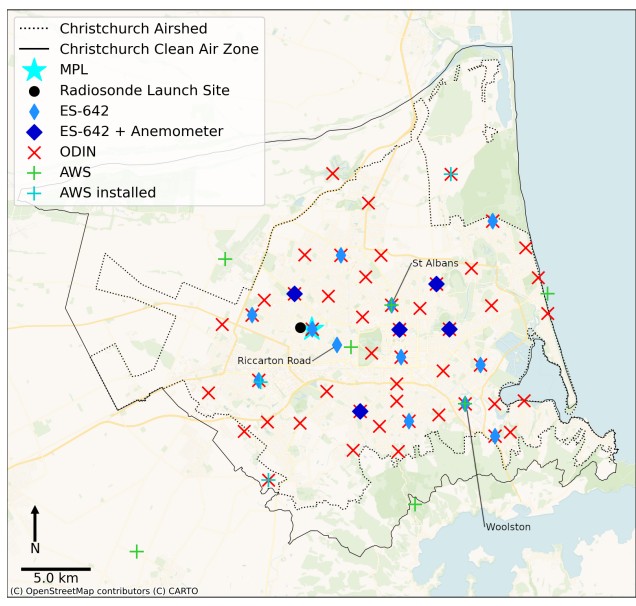

**Figure 2.** Locations of the instruments deployed during the MAPM field campaign and the AWSs operated by MetService and NIWA. The solid black line indicates the boundary of the Christchurch Clean Air Zone, the black dotted line indicates the boundary of the Christchurch Airshed. Locations of AWSs operated by members of the public are not shown. © OpenStreetMap contributors 2020. Distributed under a Creative Commons BY-SA License.

running in Christchurch during the winter 2019 field campaign. As ES-642s require a mains power supply, most of them were installed in private residential properties owned by volunteers and instruments were generally mounted onto available struc-
tures such as fence posts (Fig. 3) at a height of around 2 m above the ground. Measurements were made at 1-second intervals and are then averaged to one minute resolution by the internal software.

## 2.2 Outdoor Dust Information Node (ODIN)

ODINs are low cost nephelometers that measure concentrations of $PM_1$, $PM_{2.5}$ and $PM_{10}$ using readily available components. Each ODIN instrument consists of a plantower PMS3003 laser PM sensor and a SHT30 temperature and relative humidity
sensor regulated by a microcontroller that logs data to a Secure Digital (SD) memory card. The PMS3003 dust sensor operates by using a laser with a wavelength of $650 \pm 10$ nm) to illuminate the air sample and a light detector to measure the scattering intensity at a 90 degrees angle (Kelly et al., 2017). Unfortunately, the manufacturer does not provide information about the implementation of the Mie scattering theory to estimate the particle size distribution. Although automatic data transmission can be enabled, this functionality was not used during the MAPM field campaign to improve instrument reliability. Instead,
data were periodically retrieved from the SD card. Power is drawn from an on-board battery that is charged by a small solar panel, allowing for units to be installed in remote locations, independent of a power source.

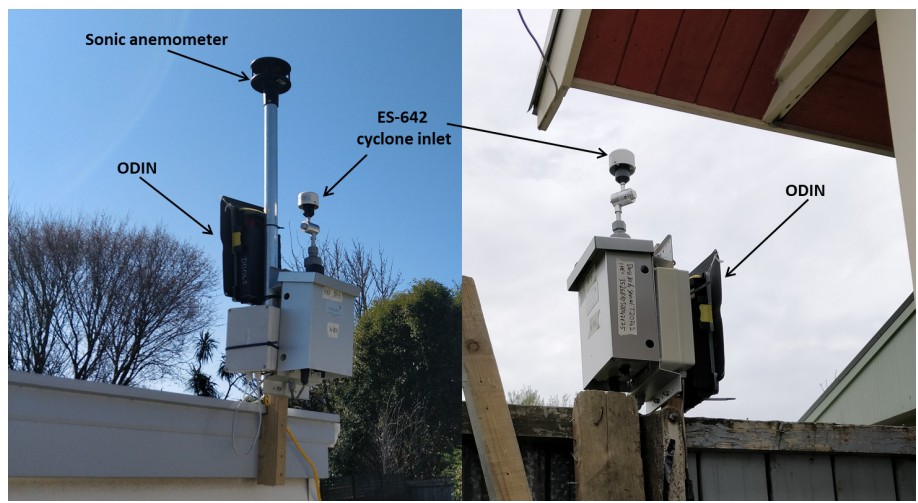

**Figure 3.** Examples of typical co-located ES-642 and ODIN installs. Note the instrument on the left is a Dust Met Mote with an additional sonic anemometer. An ODIN was co-located with every ES-642 instrument for intercomparison purposes.

Of the 50 ODINs that were deployed for the MAPM field campaign, 16 were co-located with the ES-642 instruments (one ES-642 site was deemed not suitable for a solar powered ODIN). The remaining instruments were installed throughout the city attached to light-posts (Fig. 4). Instruments were intended to be installed 2.5 m on the light-posts, however at several sites instruments were installed at a different height due to other fittings on the pole. This led to the ODIN install heights varying from 2 to 3 m Data from two ODINs could not be retrieved as one was destroyed due to water ingress and one was presumed to be stolen from the light-post.

The ODINs took instantaneous measurements at 1-minute time intervals and reported PM values as the nearest integer constraining the accuracy provided by the ODIN. The ODINs were set to sample once every 60 seconds instead of at the beginning of every minute and because of variations in the length of the sampling run, the reporting times gradually drifted and were linearly interpolated to integer minutes following the pre-screening of data, described in Appendix A.

### 2.3 Tapered Element Oscillating Microbalance (TEOM)

Three Tapered Element Oscillating Microbalance Filter Dynamics Measurement System (TEOM-FDMS, hereafter referred to as TEOM) instruments were running in Christchurch during the MAPM field campaign as part of the permanent observing system installed by ECan and provided data at hourly resolution. The TEOM instruments were co-located with an ES-642 and an ODIN instrument at the Woolston and St Albans sites and with an ES-642 at the Riccarton Road site (Fig. 2). The TEOM continuously measures $PM_{2.5}$ and $PM_{10}$ concentrations and are classified as equivalent to gravimetric measurements by the US Environmental Protection Agency (Charron, 2004). Gravimetric measurements are based on weighing the mass of particulate matter that accumulates on a filter after air has passed through the filter over a prescribed time period, generally 24 hours. The TEOM measures PM concentration by passing air through an oscillating filter (Patashnick and Rupprecht, 1991). As PM

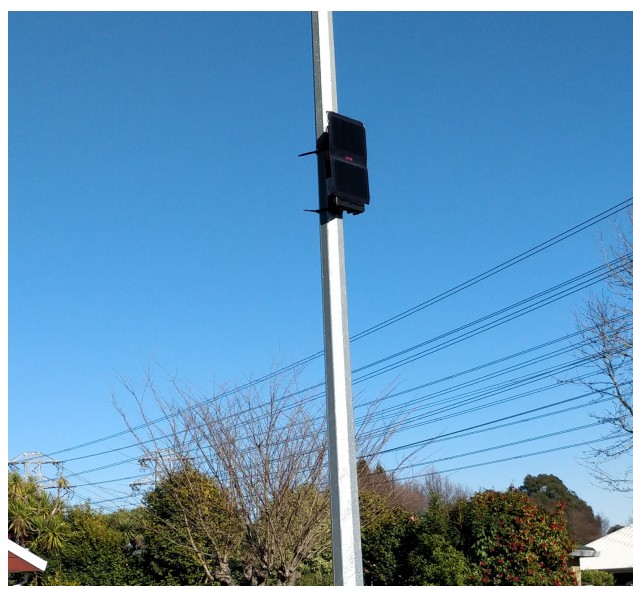

**Figure 4.** An example of a typical ODIN installation on a light-pole.

accumulates on the filter, the inertia of the filter and thus the frequency of oscillation of the filter changes. The instrument therefore measures particulate matter mass directly.

### 2.4 Automatic Weather Station (AWS)

Three temporary AWSs were installed specifically in support of the MAPM field campaign. These were deployed to supplement measurements from AWSs operated by MetService, NIWA and by members of the public who made their data available through WOW maintained by the United Kingdom Met Office. While data from all of these AWSs (a total of 30 instruments) have been used in the MAPM project, only the three dedicated MAPM AWSs will be described and here. Measurements were made using a Unidata LM34 temperature sensor, a Vector W200P Potentiometer wind vane to determine the wind direction and a Vector A101 anemometer to measure wind speed. The data were logged using a Unidata Starlogger 6004D-2, which averaged 3-second data to a 10-minute resolution and recorded the averages, the standard deviation and the minimum and maximum values measured within the preceding 10 minutes.

The instrument locations were chosen to complement the network of permanently installed AWSs. Observations at the exterior of the city were preferred to provide information on any inflow of PM across the perimeter of the city. Two AWSs were located in rural fields just outside the suburban city area, while the third was installed on an abandoned airfield towards the perimeter of the city. The instruments were installed 2 m above the local foliage (one instrument was located in a field containing a 1.5 m tall crop so was installed 3.5 m above the surface). All AWSs were installed at least 50 m from the nearest tall obstruction.

Extensive quality control was performed on all AWS data, which is described detail in Sect. 4.

## 2.5 Vertical profile measurements

The vertical stability of the atmospheric column has a strong effect on the distribution of aerosols. During night-time, radiative cooling at the surface of the atmosphere causes temperature inversions to form in the lower layers of the atmosphere. These regions of stable air prevent mixing of aerosol above the boundary layer. Therefore, to accurately simulate the transport of aerosol across a city, it is essential for any transport model to correctly represent the planetary boundary layer height (BLH). To evaluate the ability of atmospheric transport models to represent the diurnal cycle of the BLH, vertical profile measurements were made during the MAPM field using:

    i  a Sigma Space mini micro pulse lidar (miniMPL)

    ii  a Lufft CHM 15k ceilometer and

    iii  radiosondes

The miniMPL and ceilometer were ran in co-location. These instruments provided continuous profiling of the vertical structure of the atmosphere above Christchurch and were complemented during two 24-hour periods by radiosondes launched from a nearby location. BLH measurements from the MiniMPL and ceilometer are not provided but can be produced using a tool such as the Automatic Lidar and Ceilometer Framework (https://alcf-lidar.github.io/)

### 2.5.1 Mini micro pulse lidar (miniMPL)

A miniMPL was installed on the roof of the Rutherford Regional Science and Innovation Centre at the University of Canterbury (43.5225° S, 172.5841° E) at an altitude of 45 m above sea level. This building is approximately 30 m high and is surrounded by several buildings of similar height. The university campus is otherwise surrounded by a residential area of primarily single- and two-story houses. The miniMPL was installed on 17 July 2019 and operated by the University of Canterbury until the end of the MAPM field campaign.

The MiniMPL is a dual-polarisation micro pulse lidar operating at a wavelength of 532 nm at pulse repetition frequency of 2.5 kHz, with a maximum range of 30 km (Spinhirne et al., 1995; Campbell et al., 2002; Flynn et al., 2007). The MiniMPL is an aerosol backscattering lidar and a detailed description of the lidar instrument can be found in Ware et al. (2016). The MiniMPL operates similarly to other lidars and operates continuously with a temporal resolution of 2 minutes.

The instrument produces native binary files with backscatter and housekeeping meta-data, which can be converted to netCDF files using manufacturer supplied software (SigmaMPL). The measurements from this campaign have been used in Kuma et al. (2020) to demonstrate the potential of a ground-based lidar simulator for model evaluation of cloud properties. The instrument is also sensitive enough to measure aerosol backscatter on a continuous basis and can therefore be used to infer boundary layer height.

### 2.5.2 Ceilometer

A ceilometer was also installed on the roof of the Rutherford science and innovation centre next to the miniMPL (Sect. 2.5.1), pointing vertically. The ceilometer operates at an infrared wavelength of 1064 nm. The maximum range of the instrument is approximately 15 km. The instrument provides vertical profiles of backscatter with a vertical resolution of 5 m in the first 150 m and 15 m above, and a temporal resolution of 2 s. Variables such as cloud base height and planetary boundary layer height are calculated by a built-in algorithm. While the instrument was active from 1 June 2019 until the end of the MAPM field campaign, due to problems with the instruments and data transfer only and incomplete set of measurements could be retrieved from the instrument.

### 2.5.3 Radiosondes

As part of the MAPM field campaign 12 GRAW DFM-9 radiosondes were launched. The radiosonde measurements were used to identify stable inversion layers that typically form during cold and calm periods, particularly at night-time. A thermistor is used to measure the temperature with an accuracy of $\pm 0.2\,°\text{C}$ and a resolution of $\pm 0.01\,°\text{C}$ and a capacitive polymer sensor measuring relative humidity with an accuracy of $\pm 4\,\%$ and a resolution of $\pm 1\,\%$ (GRAW Radiosondes, 2019). The atmospheric pressure was calculated based on the GPS altitude of the radiosonde. Altitude, wind direction and wind speed are calculated from the Global Positioning System (GPS) location of the sonde.

Two 24 hour periods in which to launch the radiosondes were selected based on the weather conditions. In each 24 hour period six balloons were launched. The first balloon was launched at 1400 NZST (UTC + 12), followed by a launch every four hours until 1000 NZST the next day. By measuring six vertical profiles throughout the day, the depth of the boundary layer and its diurnal cycle can be investigated. Temperature inversions near the top of the boundary layer form a stable barrier preventing vertical mixing, constraining aerosol within the boundary layer. The first of two 24 hour launch periods took place on 25 of July 2019, a day that was characterised by clear, relatively cold conditions with decreasing wind speeds. Around 2200 NZST dense fog formed which evaporated around 0830 NZST the next morning. The second launch period, which began on the 15 of August 2019, was characterised by reasonably clear conditions with decreasing wind speeds towards the night and no fog occurring (Fig. 7). The primary goal of the balloon launches was to sample the air within the boundary layer. To increase the sampling rate in the boundary layer, all balloons were underinflated with a target ascent rate of $3\,\text{ms}^{-1}$ compared to the commonly used $5\,\text{ms}^{-1}$.

## 3   MAPM Field campaign design

We sought an optimal set of 50 sites around Christchurch city whose pollution measurement times series would be as different as possible from those at every other site. This design philosophy would maximise the information content of the time varying PM concentration field sampled at the 50 sites. To accomplish this we first developed a method for generating hourly spatially-resolved $PM_{2.5}$ concentration maps over the domain from point source PM measurements and model output.

## 3.1 Hourly concentration maps

The measurements used in the concentration maps were made by MOTE over the winters of 2016 and 2017 (Sect. 1.2), extreme outliers were removed and hourly averages were then calculated. We fitted a least squares regression model to every winter day over 2016, and 2017 separately using the hourly $PM_{2.5}$ measurements. The basis functions in the regression model contained spatially resolved, modelled winter maximum and winter average concentrations expanded into six Fourier terms. The modelled winter maximum and winter average of $PM_{2.5}$ concentration fields were obtained from Golders Associates

(2016), and compromised 137x137 grid cells over Christchurch. For every hour the residuals of the fits were calculated and then kriging was used to interpolate this field across the whole model domain, creating the delta map. Finally the regression model was evaluated at each grid point, and combined with the delta map, producing the gridded hourly maps of $PM_{2.5}$ concentration over Christchurch during the 2016 and 2017 winters. These maps then guided the process for locating the instruments deployed during the campaign.

## 3.2 Instrument placement

To select 50 sites for the PM instruments, we compiled a list of 32 properties of volunteers and 50,000 suitable light poles around the city to choose from. Hourly $PM_{2.5}$ concentration maps were derived from the regression model output described above at each site over June, July, and August of 2016 and 2017. In addition to these potential sites there were a number of fixed sites: i) three permanent ES-642 installations that are maintained by ECan and ii) a site at the University of Canterbury

where a ES-642 was installed to be co-located with the miniMPL. Starting with the PM concentrations of these four fixed sites, an algorithm was employed that selected the next instrument site out of the list of potential sites with the least correlation to the other sites in the set of sites already chosen. First the sites for the ES-642s were selected out of the potential sites (ODINs were also installed at all except one of the ES-642 sites), as ES-642s were only able to be installed at the volunteer sites. Secondly the sites for the remaining ODINs were selected. Because the majority of variation in the derived $PM_{2.5}$ concentration estimates

at each site were induced by the measurements made during the 2016 and 2017 campaigns (Sect. 3.1), the algorithm tended to cluster instruments close to the original measurement sites. To account for this an extra term was added to the algorithm which maximises the distance between the sites. The adjusted algorithm preferentially suggested sites on the perimeters of the city, which was desirable for estimating the background $PM_{2.5}$ concentrations flowing into the city (Fig. 2).

## 4 Quality control and correction of measurements

Overall, three versions of the $PM_{2.5}$ data sets were generated and are provided with this paper. The different versions are described in detail below, briefly:

– version 'raw': Is a collection of the measurements as obtained from the instrument but all data were put into a common netCDF file format. In addition, some pre-screening of the PM measurements was performed (see Sect. 4.1) to flag erroneous data.

- version 1.1: Contains all PM$_{2.5}$ data that were corrected to a chosen standard (see Sect. 4.2.2) to produce a consistent set of measurements, i.e. consistent between instrument types and consistent through time.

- version 2.0: As with version 1.1, this version contains all PM$_{2.5}$ data that were corrected to a chosen standard (see Sect. 4.2.2), but for version 2.0 the correction applied depends on environmental variables such as relative humidity.

In addition to the PM$_{2.5}$ data sets, netCDF files are provided for the AWS measurements, the ceilometer and MiniMPL data. While an internal consistency check was applied to the AWS data, were all 'bad' data were flagged, no screening has been performed on the ceilometer or MiniMPL data.

### 4.1 Pre-screening of the measurements

A simple pre-screening process was applied to all data from all the instruments to remove erroneous values. Firstly missing data were flagged as such, secondly a plausible range was defined for each variable and values outside this range were also flagged. The values used for these plausible ranges are listed in Appendix A. Finally other values that were clearly erroneous were flagged, for example PM$_{2.5}$ values measured by ES-642s were flagged if the air flow rate through the device fell outside the acceptable range stated on the ES-642 datasheet ($1.9 <$ flow rate $< 2.1$). For ES-642s 1.46% of PM$_{2.5}$ data points were flagged as missing and no PM$_{2.5}$ values fell outside the reasonable range ($PM_{2.5} < 10000 \, \mathrm{\mu gm^{-3}}$).

### 4.2 PM$_{2.5}$ QA/QC and correction

All PM$_{2.5}$ measurements were corrected using data collected during two co-location periods:

i  a pre-campaign co-location that ran from 6 June 2019 1700 NZST to 12 June 2019 1700 NZST

ii  a post-campaign co-location that ran from 30 August 2019 1900 NZST to 8 September 2019 1900 NZST

For both co-location periods, all PM instruments together with the TEOM instrument were located at the Woolston site ($43.5572°$ S and $172.6811°$ E). The instruments were mounted on a scaffold approximately $3 \, \mathrm{m}$ above the ground.

### 4.2.1 ODIN time retrievals

The ODIN instruments had no built in absolute reference for time. The time was set each time the instrument was installed and the instrument required constant power to the board in order to keep time. This meant that if an ODIN restarted during the campaign the time on the instrument would reset to the time that the instrument was originally started at. During the campaign ODINs restarted for a variety of reasons, presumably due to either low battery voltage (and then restarting once the solar panel recharged the battery), or due to a short on the circuit board due to ingress of debris or moisture. This resulted in several large sections of data being recorded that were unusable due to the timing of the data being unknown.

Cross correlation analysis was performed to retrieve these missing data. This retrieval method was only applied to sections of missing data containing at least 12 hours of continuous measurements. PM$_{2.5}$, temperature, and relative humidity from the

missing section of data were cross correlated, over a range of plausible times, against the median value from all operating
ODINs within $5\,\mathrm{km}$ of the instrument being corrected. The peak in the product of these three cross correlation curves was then found, if this peak was greater than 0.8 this was identified as the time offset and the section of data was corrected to match the time of this peak. Data that was retrieved using this method was flagged in the netCDF files, i.e. the flag 2 was used which is described as 'Time index retrieved using cross correlation analysis'. In total 2438 hours of data were retrieved across all ODINs.

### 4.2.2   Correcting PM$_{2.5}$ measurements

While the measurements cannot be corrected to the 'truth' as the 'true' PM$_{2.5}$ concentrations are unknown, a correction can be applied to the measurements that creates a data set that is spatially and temporally consistent. In other words, the PM$_{2.5}$ measurements can be corrected to: (i) ensure that the measurements made during the main deployment period were consistent between instruments, of either the same or different types; (ii) ensure that the measurements made during the main deployment
period by each individual instrument were consistent through time.

The correction applied to all PM$_{2.5}$ measurements is based on an approach that uses a regression model together with the PM$_{2.5}$ measurements from a chosen reference instrument. In this study all PM$_{2.5}$ measurements from the ES-642 and ODIN instruments are separately corrected to the PM$_{2.5}$ measurements from the TEOM. As the TEOM only provides hourly PM$_{2.5}$ measurements, hourly means of all valid ES-642 and ODIN measurements for each individual instrument and for each co-
location period were calculated. If fewer than 50 valid measurements are present in a given hour that hour was excluded. Furthermore, if an instrument recorded data for less than 80 % of a given co-location period the measurements were instead corrected against the other co-location period only.

Once the hourly mean concentrations have been calculated, a regression model was applied to the measurements of each ES-642 and ODIN instrument, respectively. In the first instance, we applied a regression model that is comprised of two basis
functions: (i) the PM$_{2.5}$ measurements from the respective instrument (i.e. either ES-642 or ODIN) and (ii) an offset term, viz:

$$PM_{2.5;\ TEOM} = a \times PM_{2.5;\ raw} + b \tag{1}$$

where $PM_{2.5;\ TEOM}$ are the hourly PM$_{2.5}$ concentrations measured by the reference instrument, $PM_{2.5;\ raw}$ are the hourly PM$_{2.5}$ concentrations measured by each individual instrument, and $a$ and $b$ are the fit coefficients. The regression model was applied to each co-location separately resulting in two sets of fit coefficients per instrument.

The derived fit coefficients were then used together with the measurements made during the deployment period at one minute time resolution, to obtain a corrected time-series of PM$_{2.5}$ concentrations. For each instrument a separate time-series was generated using the coefficients from each of the two co-location periods separately. These two times series were then combined using a weighted average in the form of:

$$PM_{2.5;\ corrected}(t) = x(t) \times PM_{2.5;\ coloc\ 1}(t) + y(t) \times PM_{2.5;\ coloc\ 2}(t) \tag{2}$$

where $PM_{2.5;\ corrected}(t)$ is the final corrected PM$_{2.5}$ concentration time-series, $PM_{2.5;\ coloc\ 1}(t)$ and $PM_{2.5;\ coloc\ 2}(t)$ are the times series formed when using the coefficients from the pre- and post-campaign co-location periods respectively, and $x(t)$

and $y(t)$ are the weighting coefficients which evolve linearly with time and have the following boundary conditions:

$x(t_0) = 1$

$x(t_f) = 0$

$y(t_0) = 0$

$y(t_f) = 1$

where $t_0$ and $t_f$ are the start and end of the main deployment period, respectively. This combined time-series formed the version 1 data set accompanying this study.

The regression model presented in Eq. 1 does not account for any environmental changes such as changes in humidity that
may have an impact on the measured PM$_{2.5}$ concentrations by different instruments. When looking at the differences between the TEOM measurements and the version 1 of the ODIN data (i.e. the corrected data using Eq. 1) it became apparent that the differences depend not only on the amount of PM measured but also relative humidity (see Fig. 9h). Furthermore, when looking at the PM concentrations from the TEOM versus the measurements from the ODIN or ES-642 (not shown) it became clear that the relationship is non-linear at low values of PM$_{2.5}$. As a result, we designed a second regression model that is comprised of
five basis functions in the form of:

$$PM_{2.5;\ TEOM} = a \times PM_{2.5;\ raw} + b \times PM^2_{2.5;\ raw} + c + d \times RH + e \times RH^2 \tag{3}$$

where $RH$ is the time-series of relative humidity measured by the instrument and $a$, $b$, $c$, $d$, and $e$ are the fit coefficients. The regression model described in Eq. 3 is applied in the same manner as the model described in Eq. 1, resulting in two sets of fit coefficients (one per co-location period) for each ES-642 and ODIN instrument. Applying these derived coefficients to
the measurements made during the deployment period lead to the production of a second set of corrected data; referred to as version 2.

### 4.3 Automatic weather station (AWS)

After applying coarse limit tests on each of the AWS data streams (Appendix A), measurements of

   i  air temperature

ii  relative humidity

   iii  wind speed

   iv  wind gust speed

   v  air pressure

from the 30 AWSs were tested for internal consistency. The purpose of the tests was to identify data that was recorded erro-
neously. Before conducting these internal consistency checks, for air temperature, all measurements were reduced to sea-level

temperatures assuming a moist adiabatic lapse rate of $6\ ^{\circ}\mathrm{Ckm}^{-1}$. For air pressure, the values were reduced to sea-level using the hydrostatic approximation assuming a layer mean temperature of $9.85\ ^{\circ}\mathrm{C}$. For air temperature and wind speed, comparisons between sites were challenged by some sites providing measurements as 1-minute means and other sites providing measurements as 10-minute means. As such, 10-minute 'synchronised' means were calculated for all data across all locations, i.e. means were calculated in common 10-minute blocks centred on 5, 15, 25, 35, 45 and 55 minutes past the hour.

The data are tested using a iterative method using three individual passes. On the first pass, a 'proxy' 10-minute value is estimated for each site. These proxy values are intended to be a best estimate of the value of the target variable at that site and are calculated as follows: for each AWS site, the closest other site in each of four quadrants (NE, NW, SE, SW) with a valid 10-minute mean is identified and a weighted mean (weighted by the inverse distance squared between the sites) of the four values (noting that it can be fewer than four) is then calculated. We note that these proxy values may be contaminated by erroneous data that were not excluded in the coarse data screening, but were used in the calculation of the proxy means. Therefore, on the second pass, only data that did not receive a 'D grade' in pass 1 (see below), were used to calculate the 10-minute proxy values. On the third pass, only data that did not receive a 'D grade' in passes 1 or 2, were used to calculate the 10-minute proxy values.

On each pass, differences between 10-minute means and their associated 10-minute proxies are calculated. An example of a histogram of these differences for air temperature is shown with selected percentiles and their associated 'grading' (A, B, C, or D) in Fig. 5. Each 10-minute mean receives an A, B, C, or D grade depending on the difference from its associated 10-minute proxy value in the context of the distribution shown in Fig. 5. Each measurement in the associated 10-minute time interval receives that grade. On the second pass, the 10-minute proxies are recalculated but now using only measurements that received an A, B, or C grade from pass 1. As in pass 1, those 10-minute proxies are used to derive new differences and a new histogram is used to give each measurement a revised grading. In this second pass we are more confident in the robustness of the proxy values as they are now less likely to be contaminated by erroneous values - indeed the histogram of absolute differences on the second pass (not shown) shows tighter limits on the A, B, and C gradings. Each measurement then receives a second A-D grading. The process is repeated a third time resulting in each measurement receiving a QA/QC label comprising three letters arising from each consistency check. For the analysis presented here, the poorest quality measurements (receiving a D grade on the third pass) are then excluded. This results in 12.5 % of the data being eliminated from each data set across all 30 sites, noting that for any single site, this could result in a majority of the data at the site not being used.

An example of the QA/QC labelling of the temperature measurements at the Belfast site (ALS1139) is shown in Fig. 6.

During the first period (upper panel, when the quality of the measurements was good, the three proxy series are almost identical and the majority of the data receive a final A grade. During the second period shown in the lower panel of Fig. 6, when the measurements were affected by hardware failures, the iterative revision of the proxy time series leads to increasingly robust QA/QC assessment of the quality of the measurements with the outliers frequently receiving a D grade (in some cases after receiving an A grade on the first pass). A similar QA/QC procedure was applied to the five variables listed above. Time series of recommended values, where the final grade was A, B, or C, are provided in the associated measurement AWS data files.

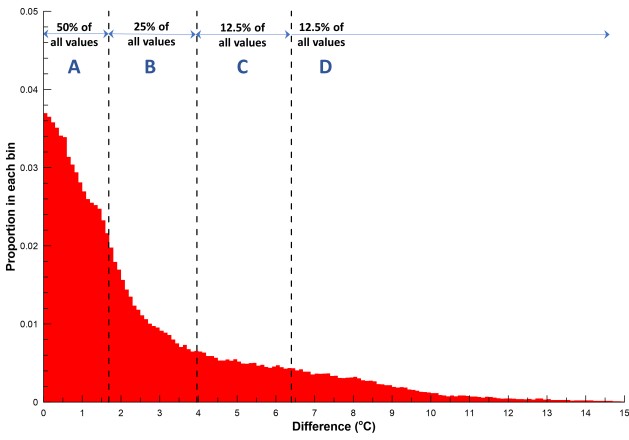

**Figure 5.** A histogram of the absolute differences between measured and proxy 10-minute air temperatures (scaled to sea-level) across all sites across the entire campaign.

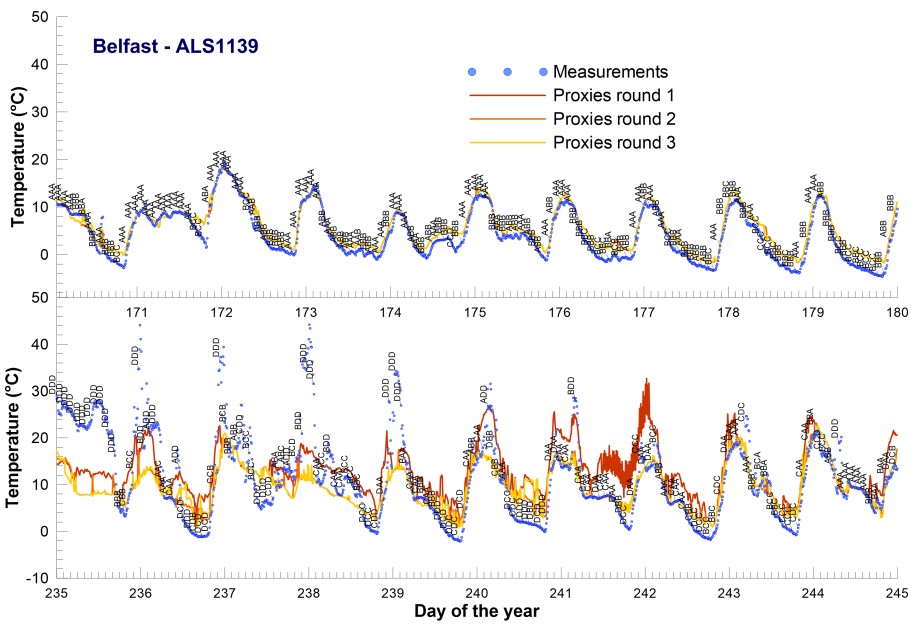

**Figure 6.** two selected periods of temperature measurements at the Belfast AWS site (ALS1139) and the QA/QC label ascribed to each of the values. For clarity, only every 10[th] label is shown. The 10-minute proxy mean time series from each of the passes (brown=1, orange=2, yellow=3) are also shown.

At three of the sites, 10-minute maximum and 10-minute minimum temperatures were also recorded. QA/QC was applied to these time series by screening out any 10-minute maximum values that were more than 5 °C above the 10-minute mean recommended value or were below the 10-minute mean. 10-minute minimum values more than 5 °C below the 10-minute

mean or above the mean were also screened out. The number of values that received a 'D' grade at each AWS are shown in
Table 1.

**Table 1.** The amount of temperature and wind speed data points that received a 'D' grade on the internal consistency check at each AWS site as a percentage of data points recorded at that site.

| Site | Temperature [%] | Wind Speed [%] |
|---|---|---|
| BDS_Belfast | 10.76 | 0.00 |
| BDS_Halswell | 3.75 | 0.00 |
| BDS_Wigram | 7.89 | 0.00 |
| Metservice_CHA | 3.00 | 0.07 |
| Metservice_CWX | 2.68 | 0.38 |
| Metservice_LBX | 21.21 | 33.41 |
| Metservice_NBX | 17.86 | 1.45 |
| Metservice_SGX | 18.68 | 39.89 |
| NIWA_Akaroa_Ews | 10.11 | 0.65 |
| NIWA_Christchurch_Kyle_St_Ews | 2.19 | 0.00 |
| NIWA_Diamond_Harbour_Ews | 8.05 | 0.34 |
| NIWA_Lincoln_Broadfield_Ews | 3.51 | 0.07 |
| NIWA_Ohoka_Cws | - | 0.00 |
| NIWA_Rangiora_Ews | 7.79 | 0.00 |
| NIWA_Waipara_West_Ews | 17.03 | 3.31 |
| NIWA_West_Eyreton_Larundel_Farm_Cws | 5.30 | 0.00 |

## 5   Uncertainties

The inverse modelling requires a quantification of the uncertainty of each measurement used. These uncertainties are used by the inverse model as an indication of how much deviation from the measurement is acceptable, in other words, how far the *measured values* are from the *true measurements*. For measurements made together with a reference instrument, the uncertainty
is simply the difference between the measurement and the reference reading. However, in order to be able to calculate these uncertainties for deployments where the reference reading is not available, we separated the uncertainty into two components: one describing the uncertainty associated to the *type* of instrument (ODIN or ES-642) and the other describing the relationship of the specific instrument to the rest of its type (inter-instrument variability). Taking this approach means that unlike the correction analysis described in section 4.2.2, measurements from a single instrument are never directly compared with the
reference instrument. The correction from section 4.2.2 creates a uniform dataset that can be analysed together, regardless of the instrument used to generate the measurement, while the uncertainty analysis estimates the differences between the measurements (raw and corrected) and a reference instrument.

Following this approach, the total uncertainty can be expressed as follows:

$$\varepsilon_x^t(m) = m - M_{\text{reference}} = (\underbrace{(\frac{1}{N_t}\sum_t m_t) - M_{\text{reference}}}_{\text{Instrument type uncertainty}}) + (\underbrace{m - \frac{1}{N_t}\sum_t m_t}_{\text{Inter-instrument variability}}) \tag{4}$$

Where

- $m$ is the measurement taken by the instrument.

- $M_{\text{reference}}$ is the reference measurement that corresponds to the measurement $m$.

- $N_t$ is the number of instruments of type $t$ that are available for this measurement.

- $\varepsilon_x^t(m)$ is the total uncertainty of measurement $m$ from instrument $x$ of type $t$, i.e., the difference between the measurement $m$ and the reference measurement $M_{\text{reference}}$.

- the *instrument type uncertainty* is the difference between the average of the measurements of all instruments of the same type $t$ and a chosen reference instrument. Here the reference instrument is the TEOM-FDMS installed at the Woolston co-location site. This uncertainty is the same for all instruments of the same type.

- the *inter-instrument variability* is the difference between the measurement $m$ and the average of measurements of instruments of type $t$ at the same time.

It is clear that this is only applicable to when a set of instruments are exposed to the same conditions, thus the two co-location periods (pre- and post-campaign) were used to calculate the uncertainty components as detailed below.

## 5.1 Data processing before analysis

The raw data from the ES-642 and ODIN instruments required some processing before they could be used to derive uncertainties. First, the flagged data were removed as described in Sect. 4.1. The remaining data were lognormally distributed so in order to use standard inferential statistics, a logarithm transformation was applied to all data to bring them within a normal distribution. This meant that any zeros or negative readings in the time-series were replaced with the detection limit of the instrument, i.e. for the ES-642s all zeros were replaced by $0.1\ \mu\mathrm{gm}^{-3}$ and for the TEOM-FDMS and the ODINs by $1\ \mu\mathrm{gm}^{-3}$.

An important difference between the two uncertainty estimates is the temporal resolution at which they can be derived. The inter-instrument variability can be derived from the native $1\ \mathrm{minute}$ resolution of the ES-642 and ODIN measurements. On the other hand, the uncertainty resulting from the instrument type can only be obtained for a time resolution compatible with that of the TEOM measurements which are available hourly. As the final output to the uncertainty calculations was a $1\ \mathrm{minute}$ time series, the hourly instrument type uncertainty was interpolated between each hour.

## 5.2 Instrument type accuracy

The first component of the measurement uncertainty corresponds to answering the question of: "*How far is the average of measurements taken by the ensemble of all instruments of the same type from that of a reference instrument*?".

Using the data from each co-location period and for each hour for which there is TEOM-FDMS data, the average of all ES-642 (or ODIN) measurements and its difference with the TEOM-FDMS reading (instrument type accuracy) were calculated. Then, a correlation analysis was performed to identify the predictive power of different variables like ambient conditions or

485 instrument readings. These analyses indicated that there was no strong correlation between the instrument type accuracy of either the ODINs or ES-642s and hourly mean temperature, relative humidity or the measured concentrations. This means that the instrument type accuracy can be added as a constant. The instrument type accuracies from pre- and post-campaign co-location data were slightly different and therefore they were interpolated over the deployment period.

It is outside of the scope of this work to fully explain and understand why the instrument type accuracy has little correlation

with ambient conditions and why their value changed between the two co-location periods. These questions will be explored in a future publication.

## 5.3 Inter-instrument variability

The second component of the measurement uncertainty corresponds to answering the question: "*How far is each device's measurement from the average of instruments of the same type?*".

Given a group of instruments of the same type sampling the same air, it is possible to define, for each instrument, the distribution of the anomalies of these measurements relative to the group's average. These distributions can be understood as the uncertainty profile of the instruments, relative to the instrument type fleet.

As both the ES642 and the ODIN units are measuring $PM_{2.5}$ every minute, a mean value and confidence interval was calculated for each type of instrument for each minute. Correlations were sought between the variability and potential environmental

factors (temperature and relative humidity) and $PM_{2.5}$ concentration.

The calculated inter-instrument variabilities showed very weak correlations with temperature or relative humidity and only the magnitude of $PM_{2.5}$ showed any predictive power for the uncertainty estimates. This is partly a reflection on the temporal resolution of the variability of the $PM_{2.5}$ measurements, which can change quickly and dramatically compared with the more gradually changing environmental factors.

For this reason, the uncertainty estimates were parameterised in terms only of the $PM_{2.5}$ for both the first and second co-locations:

$$\text{Inter-instrument variability} = \alpha * PM_{2.5} + \beta \tag{5}$$

Where $\alpha$ and $\beta$ are determined for each instrument of each type and are different from the first and second co-locations. The deployment uncertainties were estimated as a linear interpolation between those estimated using the parameters obtained from

the first co-location and those using the coefficients from the second co-location. See the code repository for the full detail of the analysis and how these terms were obtained.

It is beyond the scope of this work to explore more in detail the relationships between the uncertainty estimates and the ambient conditions which will be analysed further in a forthcoming article.

## 6   Data and analysis

Temperature and relative humidity profiles were measured on 12 radiosonde flights during the two sub-campaigns as detailed in Sect. 2. Figure 7(b-g) shows the temperature and relative humidity profiles between the ground and $1500\ \mathrm{m}$ for all launches between 1400 NZST 15 August and 1000 NZST 16 August 2019. The temperature profiles show a strong temperature inversion forming below $250\ \mathrm{m}$ as the night progresses and the surface cools radiatively. This inversion reaches its peak at 0600 NZST on 16 August (Fig. 7(f)) with a strength of $5\ ^{\circ}\mathrm{C}$.

Figure 7(a) shows the backscatter recorded by the MPL during August radiosonde launch period. Stronger backscatter is recorded near to the surface, suggesting that there is a higher concentration of aerosols in the lower atmosphere. Strong gradients in the backscatter profiles are present near regions where temperature inversions were observed by the radiosondes (shown in pink), this highlights the constraining effect that inversions have on aerosols.

The fit coefficients calculated from the pre- and post-campaign co-location periods used to correct the $PM_{2.5}$ data forming version 1 of the dataset are shown in Fig. 8. For instruments whose data was corrected against a single co-location period, due to a failure during the other co-location period, the stationary coefficient used is plotted as either a square (corrected against co-location 1) or a triangle (corrected against co-location 2). The $a$ fit coefficients (Fig. 8a) decreased from the first co-location to the second for all instruments except one. Similarly, the $b$ fit coefficients decreased for all ODINs (Fig. 8b; red) and increased slightly for all ES-642s (blue). These coefficient drifts are likely due to the differing conditions that occurred during the two co-location periods. The two co-location periods occurred at different times of the year, the PM sources would differ at these times due to seasonality of natural sources as well as differences in human activity. The synoptic time scale weather patterns that occurred during the co-locations would also have an effect on the sources of PM at the co-location site. Differing PM sources will change the size distribution and chemical make-up of the PM which may result in a change of the sensitivity of the sensor. Huggard et al. (2019) showed that although the fit did improve as the amount of the training data was increased, when training a regression model between ODIN data and TEOM data, increasing the training period from 7 to 14 days only reduced the mean squared error (MSE) by 3.8 %. This gain is minimal considering that it requires the sacrifice of valuable deployment period data. Huggard et al. (2019) also found that some time periods produced anomalous calibration values. Because of this we recommended that for future campaigns data are corrected using a series of short co-locations. If weather patterns present during the co-locations are anomalous for the given season, the co-location should be repeated as it may not be a fair representation of the seasonal PM emissions that are to be measured.

With the exception of one ES-642, all ES-642s generally showed a smaller change in magnitude of both coefficients between the two co-locations. ES-642s are able to heat incoming air, preventing the relative humidity of the incoming air exceeding

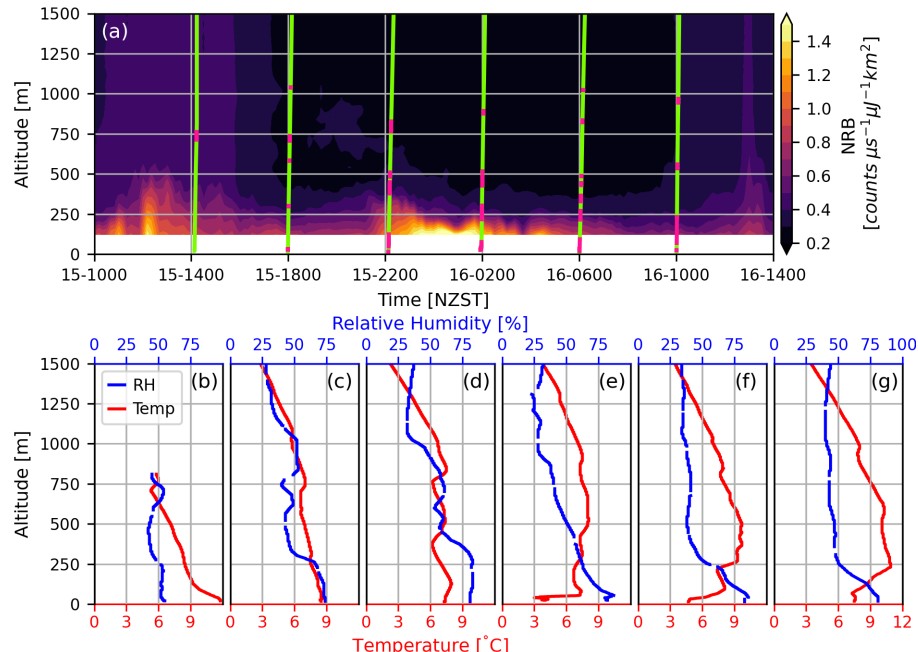

**Figure 7.** (*a*) Normalised relative backscatter (NRB) curtain taken by the miniMPL between 1000 NZST 15 August and 1400 NZST 16 August 2019, the dashed, green lines indicate the timing of the six radiosondes launched in this period with temperature inversions highlighted in pink. (*b-g*) Relative humidity and temperature profiles measured with GRAW DFM-9 radiosondes during the same period, shown in chronological order.

**Table 2.** Mean squared error (in $\mu g^2 m^{-6}$) between hourly ODIN or ES-642 data (for all three data versions) and data from the co-located TEOM at the St Albans sites using measurements made during the entire deployment period. The instrument ID for the ODIN instrument is 'SD0025' and for the ES-642 instrument the ID is 'ES_SA'.

|        | raw   | Version 1 | Version 2 |
|--------|-------|-----------|-----------|
| ODIN   | 48.81 | 32.29     | 24.96     |
| ES-642 | 30.85 | 14.75     | 19.31     |

40 %. This reduces the errors caused by the misidentification of water vapour as PM. ES-642s also used sharp-cut cyclones to prevent PM greater than $2.5\ \mu m$ entering the sensor. These factors mean that ES-642s are less susceptible than ODINs to environmental changes such as changes in humidity or particle size distribution. This is likely the reason why the change in fit coefficients, from the pre- to post-campaign co-location, for the ES-642s is smaller than that for ODINs.

A comparison of the differences between the raw, version 1, and version 2 data for the ODIN and ES-642 instruments that were co-located at the St Albans site and the St Albans TEOM ($43.5113°$ S, $172.6337°$ E; note this is a different TEOM than the instrument that the corrections were made against) and the dependence of the differences on the temperature and relative humidity measured by the instrument are shown in Fig. 9 and 10. Table 2 presents the MSE between hourly averages of the

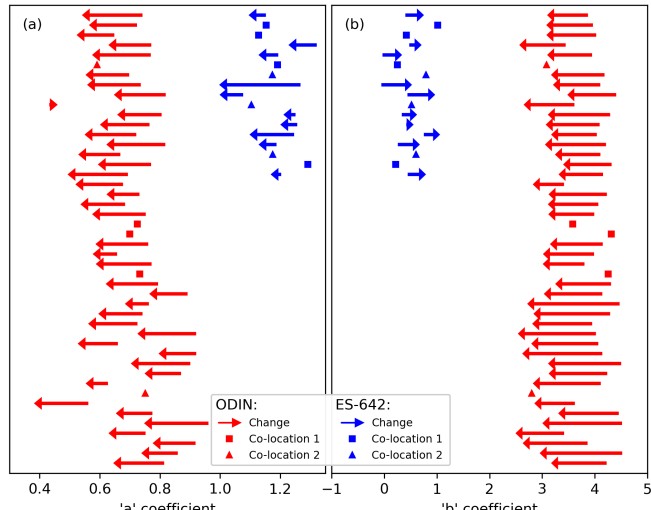

**Figure 8.** The change in the $a$ and $b$ coefficients used to generate version 1 of the data set. The fit coefficients are unit-less. The ends of each arrow indicate the $a$ and $b$ coefficients calculated at co-location 1 (tail) and co-location 2 (head) for a single ODIN (red) or ES-642 (blue). In cases where an instrument was corrected against single co-location this value is plotted as a square (co-location 1) or triangle (co-location 2).

ODIN or ES-642 data and the St Albans TEOM. Further comparison of these data sets are shown in Fig. 9 (ODIN) and Fig. 10 (ES642). The best agreement between the ODIN and the TEOM occurred with the version 2 correction (Table 2). ODINs do not have a built in mechanism to reduce uncertainty resulting from water, which causes particles to aggregate and increase in size. The uncertainty of ODIN measurements is therefore increased during periods of high ambient relative humidity (Fig. 9g-i).

The version 2 correction includes a correction based on relative humidity; this is, in part an explanation for why the version 2 performed better. The mean bias between the raw ODIN data and the TEOM at St Albans is $0.42\ \mu g^{-3}$ (Fig. 9a) this is less than that of the version 2 (Fig. 9c). However, the mean of the raw data differs significantly from the mode of the distribution and the bias shows strong asymmetry in its distribution.

While the mean bias does not appear the depend on temperature, the variance on the bias, and therefore the uncertainty

of the measurements made with this ODIN, increases at lower temperatures (Fig. 9d-f). Similarly, the variance in $PM_{2.5}$ bias increases when the relative humidity exceeds 80 %. These two trends may be related, as the relative humidity will generally increase as air cools.

In contrast, the ES-642s performed best when corrected using the simpler version 1 correction (Table 2). The version 2 performed worse than version 1 but was still an improvement on the raw data set. This suggests that the additional fit coefficients

added for version 2 resulted in over-fitting when applied to ES-642 data. Figure 10a-c shows that the ES-642 bias distributions are much more symmetrical than that of the ODIN and have a smaller standard deviation. Similar to the ODIN, the variance of the bias increases as temperature decreases, but to a lesser degree. The relation between bias and relative humidity is very different from that of the ODIN due to the inlet heater, built into an ES-642. This is likely the reason why the version 2

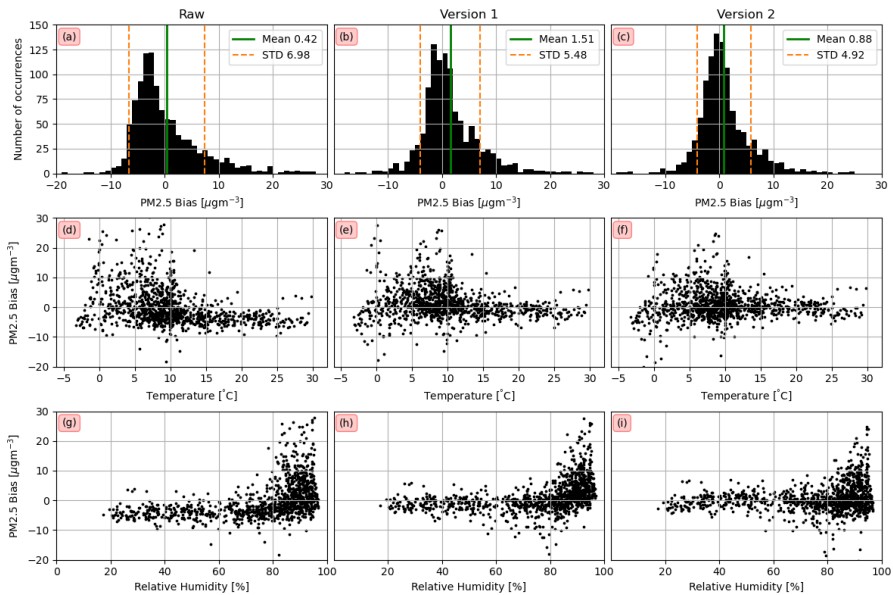

**Figure 9.** A comparison of hourly means of the raw (*a,d,g*), version 1 (*b,e,h*), and version 2 (*c,f,i*) data from ODIN and the TEOM at the St Albans site. (*a,b,c*) show histograms of bias (ODIN-TEOM) with the mean (green line) and ±1 standard deviation (orange dashes) indicated. (*d,e,f*) show scatterplots of the bias against temperature and (*g,h,i*) show scatterplots of bias against humidity. The instrument ID for the ODIN instrument is 'SD0025'.

correction performed poorly on ES-642 data compared to the simpler version 1 correction, a correction based on relative humidity was not necessary as the inlet heater prevented these biases.

The ODIN instruments measured both $PM_{2.5}$ and $PM_{10}$. Although the goal of the campaign was to measure $PM_{2.5}$, the $PM_{10}$ data were used as a diagnostic tool for the $PM_{2.5}$ measurements. We define the dimensionless value $R$ as the ratio of $PM_{2.5}/PM_{10}$. In Fig. 11, $R$ derived from measurements at two ODIN sites is compared: ODIN 172, a site near the centre of the city (Fig. 11b; $43.517°$ S, $172.615°$ E) and ODIN 156, a site on the eastern coastline (Fig. 11d; $43.498°$ S, $172.728°$ E). The distribution of calculated $R$ values measured at these sites was divided into four histograms based on the wind direction at nearby AWS stations: the Kyle street AWS (Fig. 11a; $43.531°$ S, $172.608°$ E) and the New Brighton Pier AWS (Fig. 11c; $43.506°$ S, $172.734°$ E). The histograms of $R$ for the city centre site (ODIN 172; Fig. 11b) show that under all wind directions the distribution of $R$ had a mode of approximately $0.8$ with values of $R$ rarely falling below $0.6$. This indicates that the majority of particles smaller than 10 µm were measured to also be smaller than 2.5 µm. PM sources such as home heating and transport primarily produce particles smaller than 2.5 µm. The histograms of $R$ for the coastal site (ODIN 172; Fig. 11d) show that $R$ has large variations that are dependent on the wind direction. During periods of westerly, offshore winds (red and green) the $R$ distributions closely resemble to those at the city centre site with modes of approximately $R = 0.8$. However, during periods of easterly, onshore wind (blue and orange) the distribution of $R$ has a mode of approximately $0.45$ with $R$ exceeding $0.6$ less than $10.0$ % of the time. This is consistent with a population of larger particles, primarily made up of natural sea-salt, entering

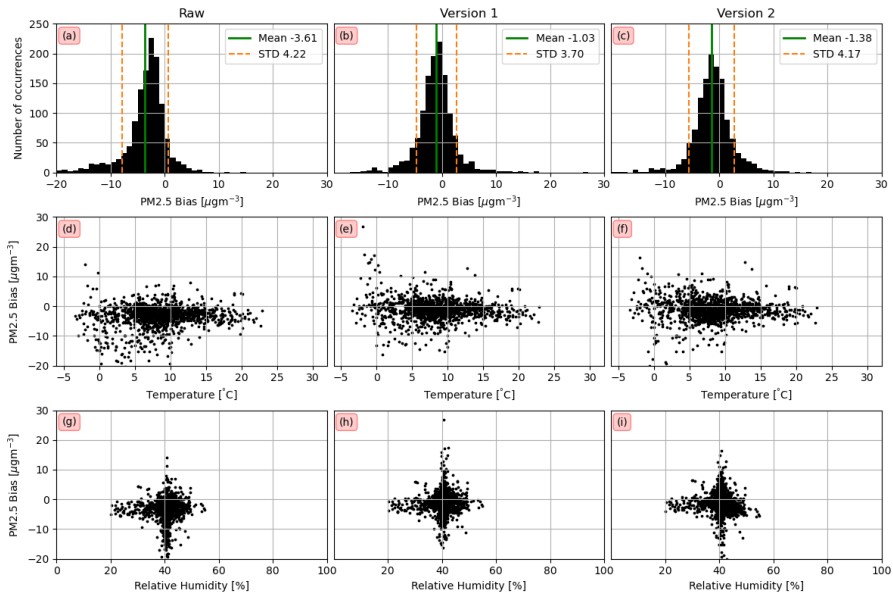

**Figure 10.** A comparison of hourly means of the raw (*a,d,g*), version 1 (*b,e,h*) and version 2 (*c,f,i*), data from the ES-642 and the TEOM at the St Albans site. (*a,b,c*) show histograms of bias (ES-642-TEOM) with the mean (green line) and ±1 standard deviation (orange dashes) indicated. (*d,e,f*) show scatterplots of the bias against temperature and (*g,h,i*) show scatterplots of bias against humidity. The instrument ID for the ES-642 instrument the ID is 'ES_SA'.

the city from the ocean. ODIN 172 was $9.36\,\mathrm{km}$ at $257°$ from ODIN 156. Although the distance between these sites was small the inland site rarely saw values of $R$ smaller than $0.6$. This highlights the increased rate of deposition that occurs in larger particles compared to smaller ($< 2.5\,\mathrm{\mu m}$) particles.

## 7   Summary

The MAPM field campaign, which ran over the winter of 2019 in Christchurch New Zealand collected variety of meteorological
and PM measurements to improve our understanding of air pollution and its distribution throughout the city. Alongside PM measurements from three types of PM instruments, three AWSs were installed to complement the 27 AWSs permanently installed in Christchurch. In addition, a mini-MPL and ceilometer were installed to provide vertical profiles of the atmosphere, and two days with 4-hourly radiosonde launches were conducted to provide additional information about the vertical structure of the boundary layer. We compare two correction methods for PM measurements, we find that the low-cost ODIN instruments
benefit from a correction that corrects based on relative humidity. We also developed uncertainties on the PM measurements. These uncertainties were separated into two components, inter-device variability and device type accuracy. The device type accuracy was found to have little dependence on environmental factors and constant values for each co-location were obtained. On the other hand the inter-instrument variability was found to vary with environmental factors. $PM_{2.5}$ and $PM_{10}$ measurements

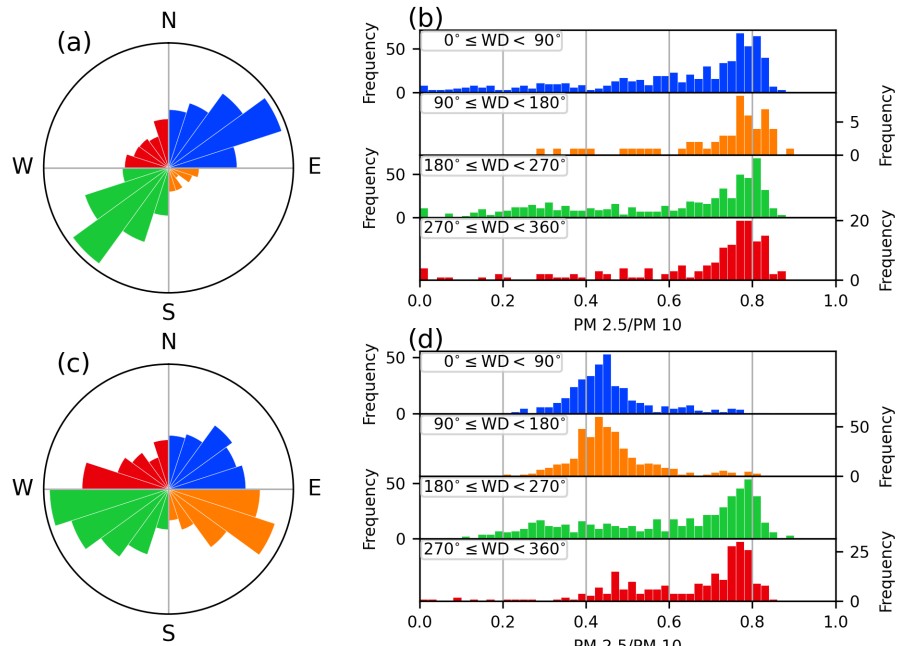

**Figure 11.** A comparison of $R$ derived from hourly mean measurements by two ODIN sites under different wind directions. (a) An angular histogram of hourly wind mean direction measured by the Kyle Street AWS, the colours indicate the quadrants used in panel b. The 'bars' are scaled for area rather than length. (b) Histograms of the $R$ derived from measurements made with ODIN 172. The data are split into four histograms based on the wind direction in panel a. Panels c and d As for a and b but instead using the New Brighton Pier AWS for the wind direction and ODIN 156 for the PM values used to calculate $R$.

at two sites, one on the coast and one near the city centre were compared. PM originating from the city was found to have a
smaller mean size than PM originating from the ocean. This methodology could be used to separate different sources of PM and identify natural and anthropogenic sources of PM. While the ES-642s outperformed the low-cost ODINs, the corrected ODIN data were found to outperform the uncorrected ES-642s. This suggests that although they are inferior instruments there is value in these low-cost sensors, particularly in situations where a high spatial resolution is desirable.

*Code availability.* Code used to calculate the uncertainties for the PM data is available at: https://github.com/bodekerscientific/MAPM_
shared

*Data availability.* The PM data collected during the campaign are publicly available from https://doi.org/10.5281/zenodo.4542559 (Dale et al., 2020b), the data from other instruments are available from https://doi.org/10.5281/zenodo.4536640 (Dale et al., 2020a). AWS data that

were collected by the permanently installed AWSs are available from NIWA (https://cliflo.niwa.co.nz/) and the United Kingdom Met Office (https://wow.metoffice.gov.uk/). The TEOM data are available on request from ECan (https://www.ecan.govt.nz/).

 **Appendix A:  Thresholds for pre-screening of data**

| Variable (formal name) | Units | Instrument(s) | Lower limit | Upper limit |
|---|---|---|---|---|
| PM$_{2.5}$ concentration | µgm$^{-3}$ | ES642, ODIN | 0 | 10000 |
| Air temperature | K | AWS, ODIN, ES-642 | 253.15 (-20 °C) | 323.15 (50 °C) |
| Air temperature | K | Radiosonde | 173.15 (-100 °C) | 293.15 (20 °C) |
| Relative humidity | % | ODIN, ES-642, Radiosonde | 0 | 100 |
| Air Pressure | hPa | ES-642 | 700 | 1300 |
| Air Pressure | hPa | Radiosonde | 0 | 1050 |
| Air flow rate | lmin$^{-1}$ | ES-642 | 0 | 10 |
| Wind speed | ms$^{-1}$ | Radiosonde | 0 | 120 |
| Wind direction | degree | Radiosonde | 0 | 360 |
| Altitude | m | Radiosonde | 0 | 35,000 |
| Geopotential height | m | Radiosonde | 0 | 35,000 |
| Latitude | degree north | Radiosonde | -90 | 90 |
| Longitude | degree east | Radiosonde | -180 | +180 |
| Dew point temperature | K | Radiosonde | 173.15 (-100 °C) | 293.15 (20 °C) |
| Virtual temperature | K | Radiosonde | 173.15 (-100 °C) | 293.15 (20 °C) |
| Ascent speed | ms$^{-1}$ | Radiosonde | -1 | 5 |
| Elevation angle | degree | Radiosonde | 0 | 90 |
| Platform azimuth angle | degree | Radiosonde | 0 | 360 |
| Horizontal range | m | Radiosonde | 0 | 300,000 |
| Air density | kgm$^{-3}$ | Radiosonde | 0 | 1.3 |

**Appendix B:  List of instruments and locations**

Table B1: The IDs and locations of the PM sensors and AWSs installed during the campaign.

| Type | Instrument ID | Latitude | Longitude | Altitude | Inlet Height |
|---|---|---|---|---|---|
| ES-642 | DM1 | -43.4880 | 172.6013 | 31.0 | 2.72 |
| ES-642 | DM2 | -43.5462 | 172.5484 | 38.0 | 3.8 |
| ES-642 | DM2 | -43.5758 | 172.5646 | 31.0 | 3.94 |
| ES-642 | DM3 | -43.5158 | 172.5441 | 41.0 | 2.68 |
| ES-642 | DM4 | -43.5354 | 172.6399 | 44.0 | 3.49 |
| ES-642 | DM5 | -43.4722 | 172.6988 | 25.0 | 2.9 |
| ES-642 | DM6 | -43.5654 | 172.6449 | 21.0 | 2.41 |
| ES-642 | DM7 | -43.5723 | 172.7004 | 20.0 | 2.52 |
| ES-642 | DM8 | -43.5225 | 172.5824 | 60.0 | 2.75 |
| ES-642 | DM9 | -43.5391 | 172.6909 | 19.0 | 1.86 |
| ES-642 | DMM2 | -43.5015 | 172.6626 | 19.0 | 2.72 |
| ES-642 | DMM3 | -43.5497 | 172.6390 | 25.0 | 2.87 |
| ES-642 | DMM4 | -43.5059 | 172.5713 | 37.0 | 3.56 |
| ES-642 | DMM5 | -43.5607 | 172.6137 | 27.0 | 2.72 |
| ES-642 | DMM6 | -43.5224 | 172.6710 | 18.0 | 2.8 |
| ES-642 | ES_RR | -43.5298 | 172.5987 | | |
| ES-642 | ES_SA | -43.5113 | 172.6337 | 12.0 | 3.35 |
| ES-642 | ES_WS | -43.5572 | 172.6811 | 8.0 | 3.56 |

| | | | | | |
|---|---|---|---|---|---|
| ODIN | SD0006 | -43.5014 | 172.6625 | 16.0 | 2.45 |
| ODIN | SD0007 | -43.5089 | 172.5500 | 16.0 | 3.34 |
| ODIN | SD0009 | -43.472 | 172.6987 | 13.0 | 2.24 |
| ODIN | SD0010 | -43.5677 | 172.6260 | -6.0 | 2.91 |
| ODIN | SD0012 | -43.5514 | 172.5920 | 9.0 | 3.05 |
| ODIN | SD0013 | -43.5336 | 172.6210 | 17.0 | 3.18 |
| ODIN | SD0015 | -43.5202 | 172.5250 | 30.0 | 2.95 |
| ODIN | SD0017 | -43.5758 | 172.5646 | 11.0 | 3.28 |
| ODIN | SD0020 | -43.5479 | 172.6370 | 11.0 | 2.72 |
| ODIN | SD0021 | -43.5059 | 172.5714 | 28.0 | 2.28 |
| ODIN | SD0022 | -43.5572 | 172.7000 | 3.0 | 3.03 |
| ODIN | SD0023 | -43.5159 | 172.5440 | 14.0 | 2.02 |
| ODIN | SD0024 | -43.5391 | 172.6908 | 3.0 | 1.2 |
| ODIN | SD0025 | -43.5113 | 172.6337 | 12.0 | 3.35 |
| ODIN | SD0028 | -43.5788 | 172.6090 | 9.0 | 3.03 |
| ODIN | SD0029 | -43.4844 | 172.7200 | 11.0 | 3.4 |
| ODIN | SD0030 | -43.5355 | 172.6399 | 9.0 | 2.83 |
| ODIN | SD0032 | -43.5793 | 172.6380 | 166.0 | 3.18 |
| ODIN | SD0033 | -43.5624 | 172.6640 | 6.0 | 3.18 |
| ODIN | SD0034 | -43.5557 | 172.7190 | 7.0 | 2.85 |
| ODIN | SD0039 | -43.5653 | 172.6450 | 11.0 | 1.75 |
| ODIN | SD0040 | -43.4940 | 172.6850 | 9.0 | 3.2 |
| ODIN | SD0041 | -43.4499 | 172.5960 | 12.0 | 3.06 |
| ODIN | SD0042 | -43.4980 | 172.6170 | 25.0 | 3.17 |
| ODIN | SD0043 | -43.5225 | 172.5827 | 35.0 | 2.09 |
| ODIN | SD0044 | -43.5662 | 172.5750 | 21.0 | 3.07 |
| ODIN | SD0045 | -43.4636 | 172.6190 | 113.0 | 2.96 |
| ODIN | SD0046 | -43.4502 | 172.6719 | 5.0 | 2.65 |
| ODIN | SD0047 | -43.5521 | 172.5160 | 39.0 | 3.17 |
| ODIN | SD0048 | -43.5927 | 172.5546 | 10.0 | 1.39 |
| ODIN | SD0049 | -43.5497 | 172.6390 | 18.0 | 2.21 |
| ODIN | SD0050 | -43.5559 | 172.6370 | 15.0 | 3.07 |
| ODIN | SD0051 | -43.4879 | 172.6270 | 13.0 | 3.16 |
| ODIN | SD0054 | -43.5656 | 172.5540 | 23.0 | 3.27 |
| ODIN | SD0055 | -43.4879 | 172.6012 | 20.0 | 2.06 |
| ODIN | SD0056 | -43.5572 | 172.6811 | 8.0 | 3.56 |
| ODIN | SD0057 | -43.515 | 172.7340 | 6.0 | 3.1 |
| ODIN | SD0058 | -43.5703 | 172.7100 | 7.0 | 2.91 |
| ODIN | SD0065 | -43.5224 | 172.6709 | 8.0 | 2.47 |
| ODIN | SD0066 | -43.5127 | 172.6520 | 5.0 | 3.17 |
| ODIN | SD0072 | -43.5723 | 172.7003 | 5.0 | 1.98 |
| ODIN | SD0074 | -43.5606 | 172.6137 | 4.0 | 2.06 |
| ODIN | SD0155 | -43.5114 | 172.6980 | 2.0 | 3.16 |
| ODIN | SD0156 | -43.4984 | 172.7280 | 7.0 | 3.02 |
| ODIN | SD0167 | -43.5070 | 172.5930 | 19.0 | 2.97 |
| ODIN | SD0170 | -43.5462 | 172.5484 | 22.0 | 3.14 |
| ODIN | SD0171 | -43.5701 | 172.5390 | 27.0 | 3.02 |
| ODIN | SD0172 | -43.5168 | 172.6150 | 11.0 | 3.31 |
| AWS | BDS_Wigram | -43.5927 | 172.5546 | 23.0 | |
| AWS | BDS_Halswell | -43.5472 | 172.5496 | 8.8 | |

| | | | | |
|---|---|---|---|---|
| AWS | BDS_Belfast | -43.4502 | 172.6719 | 0.4 |
| AWS | Metservice_CHA | -43.4890 | 172.5280 | 37.0 |
| AWS | Metservice_CWX | -43.7510 | 172.8200 | 55.0 |
| AWS | Metservice_LBX | -43.7460 | 173.1220 | 236.0 |
| AWS | Metservice_NBX | -43.5060 | 172.7340 | 9.0 |
| AWS | Metservice_SGX | -43.6040 | 172.6490 | 496.0 |
| AWS | NIWA_Akaroa_Ews | -43.8090 | 172.9660 | 45.0 |
| AWS | NIWA_Christchurch,_Kyle_St_Ews | -43.5307 | 172.6077 | 6.0 |
| AWS | NIWA_Diamond_Harbour_Ews | -43.6331 | 172.7281 | 122.0 |
| AWS | NIWA_Lincoln,_Broadfield_Ews | -43.6262 | 172.4704 | 18.0 |
| AWS | NIWA_Ohoka_Cws | -43.3423 | 172.5657 | |
| AWS | NIWA_Rangiora_Ews | -43.3286 | 172.6111 | 23.0 |
| AWS | NIWA_Waipara_West_Ews | -43.0703 | 172.6534 | 130.0 |
| AWS | NIWA_West_Eyreton,_Larundel_Farm_Cws | -43.3573 | 172.4322 | 88.0 |
| AWS | WOW_Allandale1 | -43.642 | 172.6545 | |
| AWS | WOW_Fendalton_Weather | -43.5264 | 172.5884 | |
| AWS | WOW_ICANTERB76 | -43.5217 | 172.7090 | |
| AWS | WOW_ICASHMER2 | -43.5743 | 172.6380 | |
| AWS | WOW_Ilam | -43.5156 | 172.5839 | |
| AWS | WOW_Lansdowne_Valley_Weather | -43.6147 | 172.5714 | |
| AWS | WOW_Lyttelton | -43.6008 | 172.7175 | |
| AWS | WOW_Mt_Pleasant | -43.5566 | 172.7130 | |
| AWS | WOW_Prebbleton_New_Zealand | -43.5867 | 172.5118 | |
| AWS | WOW_Rolleston | -43.6079 | 172.3677 | |
| AWS | WOW_Templeton | -43.5492 | 172.4657 | |
| AWS | WOW_Vega_Place | -43.5708 | 172.6856 | |
| AWS | WOW_West_Melton | -43.5315 | 172.3741 | |
| AWS | WOW_Worcester_Street | -43.5313 | 172.6473 | |

*Author contributions.* The field campaign was undertaken by Ethan Dale, Jordis Tradowsky, Stefanie Kremser, and Greg Bodeker with the assistance of Jonathan Barte, Jan-Niklas Schmidt, Woody Pattinson, and Nariefa Abrahim. The mini-MPL and ceilometer were installed and maintained by Adrian McDonald and Peter Kuma. Leroy Bird performed the analysis that selected instrument locations and wrote several

utility scripts to assist the post-processing of the data. The QA/QC was performed by: Ethan Dale (all PM data), Greg Bodeker (AWS data), and Jordis Tradowsky (radiosonde data). The uncertainties for the PM measurements were calculated by Gustavo Olivares, Guy Coulson, and Elizabeth Somervell. The manuscript preparation was lead by Ethan Dale with contributions made by all authors

*Competing interests.* The authors have no competing interests to declare

*Acknowledgements.* We acknowledge the New Zealand Ministry of Business, Innovation and Employment (MBIE) for funding the MAPM

project under contract BSCIF1802. We would like to thank Tim Mallet, Nathan Cross and Ben Scott from ECan for providing access to the ECan measurement site at Wollston, their help in organising the requirements for the co-locations and help in setting up all instruments

during the co-locations and deployment period. We thank Teresa Aberkane from ECan for providing us with TEOM data and offering advice on the PM corrections. We also thank Sam Edwards from NIWA who assisted in the deployment of ES-642 instruments. We thank the volunteers who hosted instruments on their private property including the Air Force Museum of New Zealand. We would like to thank the Christchurch City Council for allowing us to install ODINs on the cities light poles. We acknowledge Marwan Katurji and Laura Revell for providing insight and local knowledge on picking instrument sites. We acknowledge Metservice, Niwa, and the United Kingdom Met Office for providing AWS data from during the campaign. We also thank Hamesh Patel and Johnny Lowis for assistance during the field campaign.

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
