# Peer review of "The winter 2019 air pollution (PM2.5) measurement campaign in Christchurch, New Zealand"

_Earth System Science Data, 2020_

## Referee Comment (RC1) · Anonymous Referee #1 · 2 Dec 2020

Review of manuscript essd-2020-276 "The winter 2019 air pollution (PM2.5) measurement campaign in Christchurch, New Zealand" by Dale et al.

**General comments**

The manuscript presents data from an intensive campaign to measure spatio-temporal variations in airborne particulate matter (PM) concentrations across a large urban area in New Zealand through the winter months of 2019. PM measurements were made with 2 types of instrumentation at ~50 sites and were complemented with additional surface meteorological stations and boundary layer profiles from balloons and ground-based remote sensing. The measurement campaign aimed to generate the data required to generate hourly emissions maps using inverse methods.

Datasets of this scale are rare, especially in New Zealand where the dominant PM source (solid fuel domestic heating) differs from many cities internationally. The scale, along with the addition of atmospheric boundary measurements make this data unique. The data will undoubtedly be useful for the stated purpose as well as for direct data analysis and model validation exercises. The data is mostly well described and complete. Specific comments address areas where further description would useful, particularly regarding the methods used to calculate uncertainties and a description of fraction of data flagged by each quality control step for each instrument and location. Also, the data from the inter-comparison periods is notably absent from the datasets, so it is impossible to test the author assertions about derived calibration coefficients and uncertainties.

Regarding data quality, the data are available at the stated DOI's. Given that the authors do not intend for the 'version 2' PM correction algorithm to supersede the 'version 1" the datasets should be contained in the same file. Currently the website lists each version as an 'update' to the previous versions, so it appears as if the 'raw' and 'version 1' data are redundant. The data from the meteorological analysis are not available (for copyright reasons), neither is a list of sites used for meteorological data (only points on a map), so the readers cannot retrieve the data from the respective repositories. No analysis of which data is removed from the full AWS dataset presented, so the quality control analysis of meteorological data presented in the manuscript is largely redundant. These issues could be addressed by listing the sites used (along with metadata) in table and providing a timeseries of quality flags for each site for each 10-min period in the netCDF dataset. In addition, a much fuller description of the miniMPL and ceilometer output variables is needed for this data to be useful.

Regarding the presentation quality – the manuscript is well written but contains several sections and paragraphs that are extraneous and distract from a concise description of the data and processing (see specific comments for suggested edits). The methods are mostly well described with the exception of the method used to calculate uncertainties where a more detailed description of the procedure and equations used is needed, along with further tables and figures describing the uncertainties. The lack of a clear treatment of uncertainties is evident in the conflicting statements regarding the uncertainty in the results and conclusions.

I am sure that with some clarification of methods and addition of the inter-comparison and meteorological quality control data to the repository that the manuscript will present a make a good contribution to the scientific literature and provide a significant and useful dataset to the community.

**Specific comments (**line numbers in **bold** and suggested rewording in *italics*)

**5.** Data from only 46 to 49 ODIN instruments is presented here. Please revise.

**13-15.** "We find that for while for the ODINs a correction based on environmental conditions is beneficial, this results in over-fitting and increased uncertainties when applied to the measurements obtained using the more sophisticated ES-642s." This sentence is ambiguous, please reword (e.g. *A correction based on environmental conditions improves the quality of data retrieved from ODIN instruments but results in over-fitting and increased uncertainties when applied to the more sophisticated ES-642 instruments*)

**13-15.** This statement does not appear to be supported by analyses in the paper – both versions of the correction algorithm are shown to reduce errors with respect to the reference. No analysis of the effect of the correction on the uncertainties is presented. Please revise to be in line with results presented in the manuscript.

**57.** "data required to test and validate the MAPM methodology". Some additional context is needed here to elaborate on what the requirements for the inverse method are (e.g. spatial and temporal resolution and extent) and whether these data exist in NZ or elsewhere. This will help justify the contribution of the dataset to the literature**.**

**70.** Some discussion of similar PM field campaigns in other locations (especially internationally) would give useful context for this dataset.

**76.** "additional 10 low cost nephelometers units were". Please give details of these units in the text.

**105.** You mention the second most common wind, which begs these question – what is the most common wind? And how do these typical meteorological conditions relate to the occurrence of poor air quality?

**113-125.** It is not clear how these statements are relevant to the measurement campaign - consider removing for brevity.

**135.**  A clearer description of the campaign timeframe including designation of the 'deployment' period is needed here. Also consider changing title of section to include description of time periods for easier reference.

**138.** "i.e. around 2 to 3 m above the surface". This is quite an important piece of information – can a description of the variation in height be given?

**170.**  A short description of how the ODIN determines concentrations in different size fractions would be useful here.

**244.** No planetary boundary height data is provided in files. Please revise and/or provide the reader with links to software than can be used to derive PBL height.

**245.** Data is only provided till mid-July – please revise text or provide data for the whole campaign online.

**297.** It would be useful if the authors refer specifically to the structure of different versions of the datasets stored online and the various flags contained within them. e.g. " three version of the PM data were produced, the first "raw" version has pre-screening applied. Two corrected versions…" This would guide the reader and remove ambiguities in nomenclature between the paper and dataset.

**303.** please describe how much data from each instrument was flagged as missing or out of range. A table or some summary statistics for each instrument type would be very useful context for readers evaluating the dataset for their intended future uses.

**310-334** – you do not present the data from the smoke barrel tests or use it in corrections, so I would suggest removing this section.

**340**. "several large sections of data" – please be more specific and provide a summary of how much data from how many sites became unusable because of timing errors.

**347.** "was flagged in the netCDF files as such". Please be more specific – what is the flag called.

**376.** "These steps were repeated…". what was the motivation for introducing a new basis function? perhaps better to introduce both together, rather than the second appearing as an afterthought. At present it reads like an additional step required for the version 1 processing, rather than a separate method.

**377.** Please give some more explanation of why these particular formulations were chosen. i.e. what is the justification for a polynomial term for PM2.5?

**413.** How do you know that 3 passes is sufficient or that 12.5 % of the data is an appropriate level of data to remove? Based on the data presented in Figure 5, there is no obvious change in the histogram of differences at 87.5% that would suggest a break between good and poor quality data?

**416.** A table describing the sites used along with how much data has been removed from each site would be a useful as a reference here.

**424.** "recommended values' what do you mean here? are you referring to the proxy time series? Or do you mean the time periods recommended for use? Also, do you provide these for just the three sites you operated or all 30 sites? Please revise for clarity.

**444.** "in the absence of further co-location data' did you consider using the co-located ES-672 and ODIN? Especially given that i) corrections for ODIN instruments had similar movement between the two co-location periods, ii) the ES-462 had fairly stable corrections. These would provide timeseries of ODIN correction against the ES-642, which would allow you to test your assumption that a change in emissions and/or environmental conditions were driving these differences. Perhaps by accounting for changes in the corrections on a daily or hourly basis would reduce the ODIN errors shown in Table 1.

**466** "very weak correlations with temperature or relative humidity". Figure 10 appears to be at odds with this statement – showing clearly larger scatter with higher RH, implying that there is larger uncertainty at higher RH. This seems a critical aspect of the uncertainty given the environment that the instruments are being deployed in. Please present and discuss the analyses used to justify the uncertainty parameterisation here. At the very least the final equations used are needed.

**467** "uncertainty estimates were parameterised' – please provide the equations used to parameterise the error here.

**482.** "There was no strong correlation in the instrument type accuracy of either the ODINs or ES-642s with either hourly mean temperature or relative humidity". Again Figure 10 shows larger scatter with higher RH - this seems at odds with your statement here. Please explain.

**492.** "temperature inversion forming below 250 m". Is there a reason that the celiometer measurements and retrieval of PBL height was not presented? You note earlier it is a critical

element, and as you only have two nights of validation, it would be worth presenting this information here.

**493.** Some comment on the miniMPL backscatter measurements needed here as they are not mentioned in the text. It appears that the concentration is increasing towards the ground, so perhaps there are higher concentrations that are not being captured by the miniMPL that you could comment on?

**499-513.** This section is an application of the data and would be better suited to the end of the manuscript after the corrections and uncertainties have been discussed.

**520.** If the coefficients are changing because emission sources are changing with the season then how does this impact the ability of the measurements to infer emissions sources? Also how do you know that a linear fit between the two periods is robust? Have you tried withholding part of the co-location timeseries from each co-location to determine the quality of the correction without including the training data in the results?

**536.** The change in ODIN correction does raise concerns about the ability of the ODINS to report absolute values of PM, particularly if the emission sources varies. This may not be an issue if the objective is the measure the relative concentrations of PM from a similar source under similar environmental conditions (which seems to be the stated purpose), but this should be explicitly discussed here.

**Figure 9.** Coefficients from only 46 ODINS are presented here – please provide more detail on why units have been excluded.

**537.** Why are uncertainties calculated from the inter-comparison not presented here?

**555.** Overfitting and increased uncertainties are mentioned in the abstract, but it is not clear what results support this statement - please explain?

**557** "The relation between bias and relative humidity is very different from that of the ODIN due to the inlet heater, built into an ES-642." Could this be why the version 2 is worse than version 1? Worth discussing.

**566** "The intra-instrument variability was found to found the have little dependence on environmental factors and a constant value was used" This does not agree with 468 "[Intra-instrument variability was] parameterised in terms only of the PM2.5" - please revise.

**567** "On the other hand the instrument type accuracy was found to vary with environmental factors." This contradicts statements at line 481 "There was no strong correlation in the instrument type accuracy of either the ODINs or ES-642s with either hourly mean temperature or relative humidity, nor was there any correlation of the uncertainty estimates with higher measured concentrations." Please revise.

In order to assess the overall data quality without having to download and process the entire dataset, it would be very useful to include tables containing: Lat/lon, elevation, sensor height, fraction of data with missing flag/out of range flag/interpolated time stamp etc, for both the PM data and the AWS data.

**Dataset comments**

ODIN/ES-642 netCDF:

- The intercomparison data should be included
- The paper states 50 ODIN were deployed, but data for only 49, 47 and 46 instruments is provided for the raw, version 1 and version 2 datasets, respectively. The manuscript should state this along with the reasons that sites were excluded in the processing.
- For version 1, no coefficients are given for ODIN-SD0167
- Please state in the netCDF which geoid was used as a reference for elevations (e.g. NZGeoid2016), or are the heights given with respect to the reference ellipsoid from the GPS (i.e. GRS1980).
- Please correct the units for inlet height, which appear to be in mm.
- The use of a 16-bit integer to represent the quality control flags makes this data hard to use in many programs. Given that there are only 2 or 3 flags used, why not just store the flags separately as Boolean variables?

Meteorological data files:

- Readme files are lacking for each instrument. These should describe the naming conventions, processing steps, sources etc, including describing the meaning of 'recommended' variables.
- The AWS and radiosonde data files are fairly self-describing, but the variables in miniMPL and Ceilometer files need further description including specifying units in some cases.
- Ceilometer data is only provided till mid-July – this should be noted in the manuscript
- Why not include the proxy data in the AWS file to provide a continuous time series?
- The file is very large 8.2BG. It would be better to split this up into multiple files, perhaps by instrument type.

**Technical corrections**

**1 …**infer *airborne* particulate…

**34.** ...PM *primarily* resulting from…

**39.** "is one of the most polluted cities in New Zealand". This statement needs qualifying – e.g. *experiences poorer air quality than most cities in NZ.*

**41.** "regional councils". Need to give context for role of regional councils for readers outside of New Zealand – e.g. *the regional government responsible for managing emissions of PM*

**49.** "up to" – suggest change to *in line with*

**67-69.** this is a repeat of lines 38-40. Please revise or remove

**70.** remove comma between "sites" and "that"

**71.** "area" >> *areas*

**72.** please remove "(measuring both PM10 and PM2.5)" from parentheses.

**78.** "differences with the" >> *differences to the*

**83.** "(Sect. 3)" >> "(*refer to* Sect. 3)"

**163**. "Nine Dust Motes and five Dust Met Motes". Please use the acronyms you have defined or don't define them.

**185.** "appendix" >> *Appendix*

**189.** "and" >> *an*

**200.** as for line 163 "the Weather Observation Website (WOW) maintained by the United Kingdom Met Office." >> *WOW*

**213.** "During night-time, radiative cooling at the surface of the atmosphere causes temperature inversions to form in the lower layers of the atmosphere. These regions of stable air 215 prevent mixing of aerosol above the boundary layer. Therefore," this detail is not needed. Consider removing.

**219.** remove duplicate "mini"

**222.** "and a ceilometer" >> *and ceilometer*

**226.** as for line 163 "A Sigma Space mini micro pulse lidar" >> *The miniMPL*

**241.** as for line 163 "A Lufft CHM 15k ceilometer" >> *The ceilometer*

**248-250.**  A generic description of a radiosondes is not required. Perhaps move line 250 to the end of the following paragraph.

**269**. "This philosophy" >> "This *design* philosophy"

**280.** 'producing' >> *produced*

**296.** please add reference to Figure 2 here*.*

**459. "**question of" >> *question*

**468.**  why is deployment in italics?

**489.** "The boundary layer is of specific interest as its stability influences the concentration of pollutants such as PM2.5 at the ground level." >> consider removing this theory for brevity

**493.** "Inversion layers such as this cause the air to have a strong static stability. This prevents vertical mixing of air, constraining pollutants to the lower layer of the atmosphere. Thus, inversions play a large role in enhanced PM levels at the ground." >> consider removing this theory for brevity

**499.** "measurements as two ODIN sites is compared" >> *measurements at two ODIN sites are compared*

**Figure 7**. Is the timestamp UTC or NZST? The x-axis and caption do not agree.

**Figure 8.** Are these hourly or 1-minute averages? Please add this info to caption.

**Table 1.** The naming of the ODIN and ES-642 in the caption is not given anywhere in the manuscript so these references are confusing. Either remove names of add tables listing the names and metadata for each site. Same for captions of Figure 10 and 11.

**Table 1.** What are the units?

**537-541.** These sentences repeat – please reword.

**566** "was found to found the have" >> *was found to have*

**Appendix A** – missing negative sign for lower limit of Air Temperature for AWS

---

## Referee Comment (RC2) · Anonymous Referee #2 · 23 Dec 2020

Review ESSD-2020-276, Christchurch PM2.5

Instruments and measurement strategies well-documented. Data well ordered and easy to access via Zenodo. Good (.nc) formats. Very good intercomparison and uncertainty discussion, covering both instrument performance (precision) and reference variances (approximates accuracy).

Three (winter) month experiment, with pre- and post-experiment intercalibration periods. Actual measurement campaign perhaps 80 days but with several failures / time sync problems on ODIN sensors. Need a figure showing time periods of vaiid data for all or most sensors?

Data organized into files by location. Reader would need to extract and re-compile to reproduce e.g. Figure 8 or Figure 9. Authors could provide data extracts using their preferred tools for each or for key figures?

A bit hard to decipher symbols on Fig 2; need something larger or with greater contrast? Despite multiple attempts, I never did identify location of 17 ES-642; I also derived a different count on each of several attempts. Despite statement at line 140 that AWS were deployed at perimeter of city, I see one AWS symbol almost dead centre on the map. Dead centre = perimeter?

Line 145: "12 balloon-borne radiosondes". Sites, or sonde launches. 12 launches over three months not very helpful unless authors know from other measurements that atmosphere vertical profile remains very stable day and night and that horizontal advection remains minimal?

Line 190: Without knowing Christchurch landmarks, these sites are not obvious (and not labelled) on Figure 2.

No RH measurements or corrections on TEOM despite same humidity depended particle aggregation / disaggregation?

Lines 202 and following: 'Unidata' mentioned here refers to an instrument manufacturer rather than to the US-based weather data provisioning service https://www.unidata.ucar.edu/? Google search does NOT return link to Unidata instruments.

Lines 206 and following: Figure 2 shows two AWS at western periphery of Christchurch airshed but - as mentioned above - one dead centre on map. The dead centre location represents the park?

Line 223, period of intense radiosonde launches indicates >> 12 radiosondes?

Line 229: miniMPL data only for months 2 and 3 of this deployment?

Lines 247 and following: reader learns here of 12 radiosonde launches, every four hours on each of two 24-hour periods. Hardly a "period of intense radiosonde launches"? Two days, with 4-hour resolution, out of roughly 80 days? Many field experiments, e.g. as far back as TOGA COARE (before birth of some of these co-authors) maintained launches every 6 hours for several consecutive months. Even with slow rise rate, two days out of roughly 80 does not constitute statistically valid survey of BL height or of its diurnal (diel) variation?

Figure 8: because ordinate ranges change substantially panel b vs panel, and more in for some wind directions than others, hard to certify the discussion of lines 501 to 513.

---

## Author Comment (AC1) · 17 Feb 2021

**MAPM Campaign paper response to reviews**

**December 2020**

We would like to thank both reviewers for their helpful comments, which improved the clarity and quality of the manuscript. Point-by-point responses follow below in which the referee's comments are in blue, our response is in black.

**1 Response to reviewer 1**

**1.1 General Comments**

The manuscript presents data from an intensive campaign to measure spatio-temporal variations in airborne particulate matter (PM) concentrations across a large urban area in New Zealand through the winter months of 2019. PM measurements were made with 2 types of instrumentation at 50 sites and were complemented with additional surface meteorological stations and boundary layer profiles from balloons and ground-based remote sensing. The measurement campaign aimed to generate the data required to generate hourly emissions maps using inverse methods.

Datasets of this scale are rare, especially in New Zealand where the dominant PM source (solid fuel domestic heating) differs from many cities internationally. The scale, along with the addition of atmospheric boundary measurements make this data unique. The data will undoubtedly be useful for the stated purpose as well as for direct data analysis and model validation exercises. The data is mostly well described and complete. Specific comments address areas where further description would useful, particularly regarding the methods used to calculate uncertainties and a description of fraction of data flagged by each quality control step for each instrument and location. Also, the data from the inter-comparison periods is notably absent from the data sets, so it is impossible to test the author assertions about derived calibration coefficients and uncertainties.

Regarding data quality, the data are available at the stated DOI's. Given that the authors do not intend for the 'version 2' PM correction algorithm to supersede the 'version 1" the data sets should be contained in the same file. Currently

the website lists each version as an 'update' to the previous versions, so it appears as if the 'raw' and 'version 1' data are redundant.

We believe that this is the best way to organise the different versions, by using the Zenodo versioning system each version gains a unique DOI allowing authors to cite a specific version. A DOI for all versions also exists allowing authors to cite all versions generally. We feel that the documentation (readme files, this manuscript, and the descriptions on Zenodo) make it clear that the versions do not intend to supersede the earlier versions.

The data from the meteorological analysis are not available (for copyright reasons), neither is a list of sites used for meteorological data (only points on a map), so the readers cannot retrieve the data from the respective repositories. No analysis of which data is removed from the full AWS dataset presented, so the quality control analysis of meteorological data presented in the manuscript is largely redundant. These issues could be addressed by listing the sites used (along with metadata) in table and providing a time series of quality flags for each site for each 10-min period in the netCDF dataset.

We thank the reviewer for the suggestion and followed his advice. We have added a table that lists all AWS sites from which meteorological data were obtained during the MAPM field campaign. The table also provides some details on where to get the data from. We have also provided files in the Zenodo data set that contain the flag data for the copyrighted meteorological data.

In addition, a much fuller description of the miniMPL and ceilometer output variables is needed for this data to be useful.

These NetCDF files have been modified to be more self-explanatory.

Regarding the presentation quality – the manuscript is well written but contains several sections and paragraphs that are extraneous and distract from a concise description of the data and processing (see specific comments for suggested edits). The methods are mostly well described with the exception of the method used to calculate uncertainties where a more detailed description of the procedure and equations used is needed, along with further tables and figures describing the uncertainties. The lack of a clear treatment of uncertainties is evident in the conflicting statements regarding the uncertainty in the results and conclusions.

We addressed these comments further below.

I am sure that with some clarification of methods and addition of the intercomparison and meteorological quality control data to the repository that the manuscript will present a make a good contribution to the scientific literature and provide a significant and useful dataset to the community.

**1.2 Specific comments**

5. Data from only 46 to 49 ODIN instruments is presented here. Please revise.

A total of 50 ODINs were deployed during the field campaign and therefore we would like to state that in the abstract of the paper. However, the reviewer is correct that we only retrieved data from 49 ODINs. A data from further 3 ODINs was data retrieved was either minimal or poor quality. The reasons for this discrepancy are now described in detail in the revised manuscript (see also response further below).

13-15. "We find that for while for the ODINs a correction based on environmental conditions is beneficial, this results in over-fitting and increased uncertainties when applied to the measurements obtained using the more sophisticated ES-642s."This sentence is ambiguous, please reword (e.g. *A correction based on environmental conditions improves the quality of data retrieved from ODIN instruments but results in over-fitting and increased uncertainties when applied to the more sophisticated ES-642 instruments*)

We followed the suggestion from the reviewer and reworded the sentence to:
"We find that when compared to measurements made with a simple linear correction, a correction based on environmental conditions improves the quality of measurements retrieved from ODINs but results in over-fitting and increases the uncertainties when applied to the more sophisticated ES-642 instruments."

13-15. This statement does not appear to be supported by analyses in the paper –both versions of the correction algorithm are shown to reduce errors with respect to the reference. No analysis of the effect of the correction on the uncertainties is presented. Please revise to be inline with results presented in the manuscript.

This statement was previously ambiguous and was intended to say that the v2 correction performed better than the v1 for the ES-642s. This sentence has been edited as above.

57. "data required to test and validate the MAPM methodology". Some additional context is needed here to elaborate on what the requirements for the inverse method are (e.g. spatial and temporal resolution and extent) and whether these data exist in NZ or elsewhere. This will help justify the contribution of the dataset to the literature.

The beginning of Sect. 1.1 briefly describes the MAPM methodology and we feel that any in depth explanation about the methodology is beyond the scope of this paper and will distract from the main message of this paper. Furthermore, the paper describing the inverse model and use of the presented measurements is described in detail in Nathan et.al (2021) which has now been submitted to

ACP. However, we have clarified the need for PM 2.5 measurements for validating the method by adding (line 68/69):

"As a result, a three month measurement campaign was conducted in Christchurch in 2019, which provides the required $PM_{2.5}$ measurements that are used as input to the inverse model, which is used to infer PM emissions sources in Christchurch. This paper describes this field campaign and obtained measurements in detail. For a detailed description about the inverse model and inferred emissions maps, the reader is referred to Nathan et.al (2021)."

Nathan, B., Kremser, S., Mikaloff-Fletcher, S., Bodeker, G., Bird, L., Dale, E., Lin, D., Olivares, G., and Somervell, E.: The MAPM (Mapping Air Pollution eMissions) method for inferring particulate matter emissions maps at city-scale from in situ concentration measurements: description and demonstration of capability, Atmos. Chem. Phys. Discuss. [preprint], https://doi.org/10.5194/acp-2020-1303, in review, 2021.

70. Some discussion of similar PM field campaigns in other locations (especially internationally) would give useful context for this dataset.

The author feel that this is outside the scope of this article. Previous campaigns in Christchurch have been discussed.

76. "additional 10 low cost nephelometers units were". Please give details of these units in the text.

It is unclear what type of sensor were used in this study.

105. You mention the second most common wind, which begs these question –what is the most common wind? And how do these typical meteorological conditions relate to the occurrence of poor air quality?

As is stated on line 103 the predominant wind in Christchurch comes from the West. The effect of these flows on PM conditions is described on line 104-105.

113-125. It is not clear how these statements are relevant to the measurement campaign - consider removing for brevity.

lines 122-125 have been removed.

135. A clearer description of the campaign time frame including designation of the 'deployment' period is needed here. Also consider changing title of section to include description of time periods for easier reference.

The time period for the deployment and co-locations periods have been added

as well as clearer definitions of these terms.

138. "i.e. around 2 to 3 m above the surface". This is quite an important piece of information – can a description of the variation in height be given?

We added more information about the installation heights and why they vary in Section 2.1 and 2.2.

170. A short description of how the ODIN determines concentrations in different size fractions would be useful here.

We added a brief description to the manuscript.

244. No planetary boundary height data is provided in files. Please revise and/or provide the reader with links to software than can be used to derive PBL height.

Link to a suitable tool has been provided.

245. Data is only provided till mid-July – please revise text or provide data for the whole campaign online.

We revised the text accordingly. The instrument was operated for the whole campaign but due to problems with that data acquisition for the instrument, not all data could be retrieved.

297. It would be useful if the authors refer specifically to the structure of different versions of the datasets stored online and the various flags contained within them. e.g. " three version of the PM data were produced, the first "raw" version has pre-screening applied. Two corrected versions..." This would guide the reader and remove ambiguities in nomenclature between the paper and dataset.

We thank the reviewer for this suggestion and we followed his advice and added a paragraph to Sect. 4 in the revised manuscript that briefly outlines the different versions of the data sets.

303. please describe how much data from each instrument was flagged as missing or out of range. A table or some summary statistics for each instrument type would be very useful context for readers evaluating the dataset for their intended future uses.

Some statistics on this have been added.

310-334 – you do not present the data from the smoke barrel tests or use it in corrections, so I would suggest removing this section.

We agree with the reviewer and have removed the section about the smoke barrel tests.

340. "several large sections of data" – please be more specific and provide a summary of how much data from how many sites became unusable because of timing errors.

No sites were totally unusable, the amount of data retrieved has been added to the doc.

347."was flagged in the netCDF files as such". Please be more specific – what is the flag called."

We clarified which flag was used to mark the ODIN data that were the time stamp was retrieved based on the method described in Sect. 4.2.2 of the manuscript.

376. "These steps were repeated...". what was the motivation for introducing a new basis function? perhaps better to introduce both together, rather than the second appearing as an afterthought.At present it reads like an additional step required for the version 1 processing, rather than a separate method.

We agree with the reviewer that the structure of the section was confusing and therefore we have re-structure Sect. 4.2.3 of the manuscript and have added some additional information where appropriate.

377. Please give some more explanation of why these particular formulations were chosen. i.e. what is the justification for a polynomial term for PM2.5?

We have clarified the design of the regression model and why we choose the basis-functions as we did in the revised section of the manuscript.

**413.** How do you know that 3 passes is sufficient or that 12.5% of the data is an appropriate level of data to remove? Based on the data presented in Figure 5, there is no obvious change in the histogram of differences at 87.5% that would suggest a break between good and poor quality data?

4 passes were applied to the AWS data and it was found that less than 0.1% of grades changed from using 3 passes. From this it was decided that 3 passes provided adequate convergence. The 12.5% threshold was chosen in a qualitative manner. There were periods of obviously erroneous data that were not entirely removed until at least 10% of data were removed, as a result we choose 12.5% as our threshold.

416. A table describing the sites used along with how much data has been

removed from each site would be a useful as a reference here.

Table has been added.

424. "recommended values' what do you mean here? are you referring to the proxy time series? Or do you mean the time periods recommended for use? Also, do you provide these for just the three sites you operated or all 30 sites? Please revise for clarity.

The 'recommended values' refer to the data that we recommend to be used as they have been screened according to the flag values set. So if a user doesn't want to use the flag values, they can just load the 'recommended' data and then should have a data set where some quality control was performed. While the recommended values were calculated for all AWS sites, the recommended values are only provided on Zenodo for the 3 AWSs installed for this campaign as we can't share the meteorological data from the 3rd party AWS sites. We have clarified this in the revised manuscript.

444. "in the absence of further co-location data' did you consider using the co-located ES-672 and ODIN? Especially given that i) corrections for ODIN instruments had similar movement between the two co-location periods, ii) the ES-462 had fairly stable corrections. These would provide timeseries of ODIN correction against the ES-642, which would allow you to test your assumption that a change in emissions and/or environmental conditions were driving these differences. Perhaps by accounting for changes in the corrections on a daily or hourly basis would reduce the ODIN errors shown in Table 1.

In short, yes, we did consider it but the goal of this dataset is to inform the inverse modelling, rather than try to obtain the most accurate readings from all instruments. One of the key questions of the MAPM study is to assess what is the impact of having *inaccurate* measurements available for an inverse modelling exercise. The question of how the emissions or environmental conditions affect the readings of low-cost sensors is beyond the scope of this study but one that we look forward to tackle with this dataset and using, among others, the approach suggested.

466 "very weak correlations with temperature or relative humidity". Figure 10 appears to be at odds with this statement –showing clearly larger scatter with higher RH, implying that there is larger uncertainty at higher RH. This seems a critical aspect of the uncertainty given the environment that the instruments are being deployed in. Please present and discuss the analyses used to justify the uncertainty parameterisation here. At the very least the final equations used are needed.

Figure 10 should not be used to draw conclusions about the uncertainty estimates because, as we indicated in the revised text, the uncertainty estimates

do not compare the readings of any individual instrument against those of the reference. We followed this approach because we wanted to quantify how different the uncertainty of the two *types* of instruments were. Also, the uncertainty estimates are produced for both raw (uncorrected) and v01 (corrected) datasets and the effect of ambient conditions is specifically addressed through the corrections to the raw data.

467 "uncertainty estimates were parameterised' – please provide the equations used to parameterise the error here.

The revised text includes the parameterisation equation for the inter-instrument variability component of the uncertainty.

482. "There was no strong correlation in the instrument type accuracy of either the ODINs or ES-642s with either hourly mean temperature or relative humidity". Again Figure 10 shows larger scatter with higher RH -this seems at odds with your statement here. Please explain.

See our response above

492. "temperature inversion forming below 250 m". Is there a reason that the celiometer measurements and retrieval of PBL height was not presented? You note earlier it is a critical element, and as you only have two nights of validation, it would be worth presenting this information here.

While the reviewer is correct that boundary layer heights are important, providing a detailed analysis of the boundary layer heights derived from the radiosondes and ceilometer data is beyond the scope of this paper. Here we would like to focus on presenting the measurements, rather than any data products that can be derived from the measurements itself. The boundary layer heights are not a standard output from the celiometer or miniMPL and different methods are available for the calculation of the boundary layer heights. As a result, a detail study and analysis of the boundary layer heights will be presented in a different publication.

493. Some comment on the miniMPL backscatter measurements needed here as they are not mentioned in the text. It appears that the concentration is increasing towards the ground, so perhaps there are higher concentrations that are not being captured by the miniMPL that you could comment on?

A paragraph has been added discussing this MPL profiles and their relation to the sonde profiles.

499-513. This section is an application of the data and would be better suited to the end of the manuscript after the corrections and uncertainties have been discussed.

We agree with the reviewer and have moved this part of the manuscript to the end of the Section.

520. If the coefficients are changing because emission sources are changing with the season then how does this impact the ability of the measurements to infer emissions sources?

The PM measurements alone can not infer emissions sources. Only together with an inversion model can measurements be used to infer emissions sources. The change of the emission sources with the season will not have an effect on the ability to infer emissions as with fewer emissions the measured concentrations will be lower.

Also how do you know that a linear fit between the two periods is robust? Have you tried withholding part of the co-location time series from each co-location to determine the quality of the correction without including the training data in the results?

536. The change in ODIN correction does raise concerns about the ability of the ODINS to report absolute values of PM, particularly if the emission sources varies. This may not be an issue if the objective is the measure the relative concentrations of PM from a similar source under similar environmental conditions (which seems to be the stated purpose), but this should be explicitly discussed here.

Yes, the changes in the corrections applied, perticularly to the ODIN data, does raise questions about the detailed response of these devices to different sources of $PM_{2.5}$ but as the reviewer identified, our objective in this study was to generate a dataset for an inversion modelling exercise in an area with relatively homogeneous emission sources and stable environmental conditions. However, these datasets will be used, together with others generated elsewhere, to explore the response of low-cost sensors to different emission sources and in varying environmental conditions.

Figure 9. Coefficients from only 46 ODINS are presented here –please provide more detail on why units have been excluded.

The fit coefficients are unit-less and we have now clarified that in the caption of the Figure. We have now explained above why we only show the results from 46 ODINs.

537. Why are uncertainties calculated from the inter-comparison not presented here?

Uncertainties were not calculated for the inter-comparison periods as this data

was used to train the regression model used for the uncertainty calculation.

555. Over fitting and increased uncertainties are mentioned in the abstract, but it is not clear what results support this statement - please explain?

Detail has been added to this paragraph to explain this

557. "The relation between bias and relative humidity is very different from that of the ODIN due to the inlet heater, built into an ES-642." Could this be why the version 2 is worse than version 1? Worth discussing.

As, above sentence has been added.

566. "The intra-instrument variability was found to found the have little dependence on environmental factors and a constant value was used" This does not agree with 468 "[Intra-instrument variability was] parameterised in terms only of the PM2.5" -please revise.

See the revised text where we attempt to clarify the description of the uncertainty components

567. "On the other hand the instrument type accuracy was found to vary with environmental factors." This contradicts statements at line481"There was no strong correlation in the instrument type accuracy of either the ODINs or ES-642s with either hourly mean temperature or relative humidity, nor was there any correlation of the uncertainty estimates with higher measured concentrations."Please revise.

This was an oversight. The corrected text is now consistent between the uncertainties section and the final summary

In order to assess the overall data quality without having to download and process the entire dataset, it would be very useful to include tables containing: Lat/lon, elevation, sensor height, fraction of data with missing flag/out of range flag/interpolated time stamp etc, for both the PM data and the AWS data.

We have added a table listing all instruments and their locations to the appendix. Another table has been added listing the number of values flagged out by various flags at various AWS sites.

**1.3   Dataset comments**

**1.3.1   ODIN/ES-642 netCDF**

- The intercomparison data should be included-The paper states 50 ODIN were deployed, but data for only 49, 47 and 46 instruments is provided for the raw, version 1 and version 2 datasets, respectively. The manuscript should state this along with the reasons that sites were excluded in the processing.

  We have now added an explanation about we didn't obtain measurements from all ODINs. We are now also providing the data from the co-location periods on Zenodo for any user who might be interested in them.

- For version 1, no coefficients are given for ODIN-SD0167

  ODIN-SD0167 only recorded 1 hour of data during the deployment period so was excluded from versions 1.1 and 2.0. This file was erroneously included in version 1.1. It has now been removed.

- Please state in the netCDF which geoid was used as a reference for elevations (e.g. NZGeoid2016), or are the heights given with respect to the reference ellipsoid from the GPS (i.e. GRS1980).

  We have added this geoid to altitude description in the .nc files.

- Please correct the units for inlet height, which appear to be in mm.

  We have corrected this error.

- The use of a 16-bit integer to represent the quality control flags makes this data hard to use in many programs. Given that there are only 2 or 3 flags used, why not just store the flags separately as Boolean variables?

  Some variables, for example mean temperature measured by the AWSs have as many as 8 flags. There is no option for Boolean values within netCDF, so flags would have to be stored as shorts. This would drastically increase the data size of these files. That's why we chose to use 16-bit integer.

**1.3.2 Meteorological data files**

- Readme files are lacking for each instrument. These should describe the naming conventions, processing steps,sources etc, including describing the

meaning of 'recommended' variables.

- The AWS and radiosonde data files are fairly self-describing, but the variables in miniMPL and Ceilometer files need further description including specifying units in some cases.

These files have been updated to be better self-describing

- Ceilometer data is only provided till mid-July – this should be noted in the manuscript

this has been noted.

- Why not include the proxy data in the AWS file to provide a continuous time series?

The proxy data was never designed to be used to fill in 'gaps' within the data, and would represent lower quality data. To avoid confusion we decided to leave this out.

- The file is very large 8.2GB. It would be better to split this up into multiple files, perhaps by instrument type.

files have been seperated.

**1.4  Technical corrections**

1. ...infer *airborne* particulate...

Change has been made.

34. ...PM *primarily* resulting from...

Change has been made.

39. "is one of the most polluted cities in New Zealand". This statement needs qualifying –e.g. *experiences poorer air quality than most cities in NZ*.

Sentence has been removed following comment regarding line 67-69

41. "regional councils". Need to give context for role of regional councils for readers outside of New Zealand –e.g. the regional government responsible for managing emissions of PM

Change has been made

49. "up to" –suggest change to in line with

Change has been made

67-69. this is a repeat of lines 38-40. Please revise or remove

Sentence in line 39-40 has been removed.

70. remove comma between "sites" and "that"

Change has been made

71. "area" >> areas

Change has been made

72. please remove "(measuring both PM10 and PM2.5)" from parentheses.

Change has been made.

78."differences with the" >> differences to the

Change has been made.

83. "(Sect. 3)" >> "(refer to Sect. 3)"

Change has been made.

163. "Nine Dust Motes and five Dust Met Motes". Please use the acronyms you have defined or don't define them.

We would prefer to refer to the full description in this context to make sure the reader is aware of the different configurations we used. The acronyms were defined because they are used in the file names of the data files that accompanies the manuscript.

185. "appendix" >> Appendix

Change has been made.

189. "and" >> an

Change has been made.

200. as for line 163 "the Weather Observation Website (WOW) maintained

by the United Kingdom Met Office." >> WOW

Change has been made.

213. "During night-time, radiative cooling at the surface of the atmosphere causes temperature inversions to form in the lower layers of the atmosphere. These regions of stable air prevent mixing of aerosol above the boundary layer. Therefore," this detail is not needed. Consider removing.

We prefer to leave this statement in the manuscript as it motivates why knowing the height of the boundary layer is important.

219. remove duplicate "mini"

Change has been made.

222. "and a ceilometer" >> and ceilometer

Change has been made.

226. as for line 163 "A Sigma Space minimicro pulse lidar" >> The miniMPL

Change has been made.

241. as for line 163 "A Lufft CHM 15k ceilometer" >> The ceilometer

Change has been made.

248-250. A generic description of a radiosondes is not required. Perhaps move line 250 to the end of the following paragraph

Change has been made

269. "This philosophy" >> "This design philosophy"

Change has been made.

280.'producing' >> produced

We believe that 'producing' is the correct word to be used in this context here but we noticed that a ',' was missing which has been corrected.

296. please add reference to Figure 2 here.

Reference added.

459. "question of" >> question

Change has been made.

468. why is deployment in italics?

Italics has been removed.

489. "The boundary layer is of specific interest as its stability influences the concentration of pollutants such as PM2.5 at the ground level." >> consider removing this theory for brevity

Sentence has been removed.

493. "Inversion layers such as this cause the air to have a strong static stability. This prevents vertical mixing of air, constraining pollutants to the lower layer of the atmosphere. Thus, inversions play a large role in enhanced PM levels at the ground." >> consider removing this theory for brevity

Sentence has been removed.

499. "measurements as two ODIN sites is compared" >> measurements at two ODIN sites are compared.

Change has been made.

Figure 7. Is the timestamp UTC or NZST? The x-axis and caption do not agree.

The time on the x-axis has been corrected to NZST.

Figure 8. Are these hourly or 1-minute averages? Please add this info to caption.

Hourly and the information has been added to the figure caption.

Table 1.The naming of the ODIN and ES-642 in the caption is not given anywhere in the manuscript so these references are confusing. Either remove names of add tables listing the names and metadata for each site. Same for captions of Figure 10 and 11.

We have clarified that in the revised manuscript.

Table 1.What are the units?

The units are $\mu g^2 m^{-6}$ i.e. PM concentration squared, this has been added to the caption for clarity.

The sentences have been combined.

Change has been made.

Appendix A–missing negative sign for lower limit of Air Temperature for AWS.

This has been corrected.

**2  Response to reviewer 2**

Instruments and measurement strategies well-documented. Data well ordered and easy to access via Zenodo. Good (.nc) formats. Very good intercomparison and uncertainty discussion, covering both instrument performance (precision) and reference variances (approximates accuracy).

Three (winter) month experiment, with pre- and post-experiment intercalibration periods. Actual measurement campaign perhaps 80 days but with several failures / time sync problems on ODIN sensors. Need a figure showing time periods of vaiid data for all or most sensors?

A figure of this type was made (see fig 1). But it was not included in the manuscript as it was difficult to read due to the large number of instruments and to keep the document length down.

Data organized into files by location. Reader would need to extract and re-compile to reproduce e.g. Figure 8 or Figure 9. Authors could provide data extracts using their preferred tools for each or for key figures?

We believe that having separate files is the most straight-forward and convenient method. If all sites were provided in one file the file size would become very large and users would have to filter out the data they do not want to use.We are hesitant to provide duplicate versions of the same data to avoid causing confusion for users.

A bit hard to decipher symbols on Fig 2; need something larger or with greater contrast? Despite multiple attempts, I never did identify location of 17 ES-642; I also derived a different count on each of several attempts. Despite statement at line 140 that AWS were deployed at perimeter of city, I see one AWS symbol almost dead centre on the map. Dead centre = perimeter?

[Figure]

Figure 1: timeline of data collected form various PM instruments.

Three AWS were installed at the perimeter of city specifically for the field campaign and the other AWS sites are permanent sites that are operated by MetService or NIWA or are privately owned. We agree with the reviewer that the symbols are rather difficult to be identified and have updated the figure. Symbols are now larger and colours have been changed.

Line 145: "12 balloon-borne radiosondes". Sites, or sonde launches. 12 launches over three months not very helpful unless authors know from other measurements that atmosphere vertical profile remains very stable day and night and that horizontal advection remains minimal?

The intent of the radiosonde launches was to act as a validation method for

MPL-derived boundary layer heights, rather than as a standalone measurement.

Line 190: Without knowing Christchurch landmarks, these sites are not obvious (and not labelled) on Figure 2.

Labels have been added to figure 2.

No RH measurements or corrections on TEOM despite same humidity depended particle aggregation / disaggregation?

Correct, no corrections were made on the TEOM measurements as this is our reference instrument.

Lines 202 and following: 'Unidata' mentioned here refers to an instrument manufacturer rather than to the US-based weather data provisioning service `https://www.unidata.ucar.edu/`? Google search does NOT return link to Unidata instruments

Correct, the temperature sensors used in the AWSs are manufactured by Unidata (`https://www.unidata.com.au/`) and that is what we are referring to.

Lines 206 and following: Figure 2 shows two AWS at western periphery of Christchurch airshed but - as mentioned above - one dead centre on map. The dead centre location represents the park?

We have clarified the wording around line 206 - we are referring here to the AWSs that were installed for the measurement campaign specifically, not the AWSs that are operated by third parties. We have changed Fig. 2 so that now, these three AWSs locations can be identified more easily.

Line 223, period of intense radiosonde launches indicates >> 12 radiosondes?

We have changed this to clarify the amount of radiosondes used and am now use the following wording "... and were complemented during two 24-hour periods by radiosondes launched from a nearby location." See also our response to the second comment below.

Line 229: miniMPL data only for months 2 and 3 of this deployment?

Correct, the miniMPL was not available prior to 17 July as it had to be shipped from the US, and custom delays caused a delay in the instalment of the instrument.

Lines 247 and following: reader learns here of 12 radiosonde launches, every four hours on each of two 24-hour periods. Hardly a "period of intense radiosonde launches"? Two days, with 4-hour resolution, out of roughly 80 days?

Many field experiments, e.g. as far back as TOGA COARE (before birth of some of these co-authors) maintained launches every 6 hours for several consecutive months. Even with slow rise rate, two days out of roughly 80 does not constitute statistically valid survey of BL height or of its diurnal (diel) variation?

Okay, that is a fair point and we have changed the wording now to clarify the amount of radiosondes in line 223 which is mentioned above and also line 488 and in the summary and abstract.

Figure 8: because ordinate ranges change substantially panel b vs panel, and more in for some wind directions than others, hard to certify the discussion of lines 501 to 513.

The ordinate ranges vary because the number of occurrences within each wind direction range varies (i.e. there were very few periods where the wind came from the South-East at the Kyle Street AWS (panel A)). Since it is the relative amount of R values that matters we chose to vary the frequency ranges such that the histogram filled each set of axes to maximise readability. We feel that this figure best demonstrates the discussion in lines 501 to 513, as it is clear that in panel (b) R values of 0.8 are most frequent regardless of wind direction. We feel that adding a consistent ordinance range would make this figure more difficult to read.

---

## Author Response (AR2)

We thank the editor for providing us with feedback, our responses to the specific comment are below.

1) Fix text at lines 102, 103;

Text has been adjusted for clarity.

2) Line 152: lower, vs lowering?

Changed to reducing

3) Line 162, MOTE Ltd: instrument manufacturer, environmental consulting company?

We included the link to the company website.

4) Line 174: plantower?

Link has been added

5) Line 174: Do you need to indicate SHT sensors as based on Sensirion?

Added this, as well as link.

6) Line 208: Because I, many friends, and likely many northern hemisphere readers worked with and know US-based UNIDATA, please follow reviewer advice and include Aussie Unidata url here, at first use.

Link added.

7) Line 254: an not and?

Error has been corrected.

8) Line 280: MOTE here indicates a project, company, ministry, ?

This sentence has been adjusted for clarity.

9) Line 281: Not clear what your regression provides? You regress PM concentrations vs location? Vs met variable (T, RH)? Modeled vs measured? Site to site correlation?

The regression model was fit to the PM observations. This paragraph was adjusted for clarity.

10) Line 303: But you earlier implied that exceedence values of PM came from residential heating using solid fuel (wood, coal) sources. Quantifying / modeling PM concentrations across the city requires knowing advective inputs plus local production? Fixed sites already provided sufficient local information?

While local emissions are the primary source of PM within the city, understanding the incoming emissions from advection is important for the functionality of MAPM.

11) Line 314 - but earlier (line 255) you told us that ceilometer data were "incomplete"?

While we do not have data for the entire period we are still able to provide the data we have.

12) Line 315 - where vs were? So ceilometer only raw, not QC'd. Why not leave them out? In fact, you never discuss them. Not useful?

Typo corrected. The ceilometer data is made available as it may be of use to others.

13) Line 429 - missing ')' here?

Bracket added.

14) Line 445 - calculation of full uncertainty budgets (e.g. instrumental, operational, environmental) rarely comes out as simple as reference - measured. As subsequent discussion in this paragraph proves!

This sentence has been removed.

15) Line 495 - you mean not the same meteorological or trajectory source air, but air containing the same particle concentrations and size distributions?

We mean the latter; the air being sampled is identical in every way. The sentence has been adjusted.